# GROUNDING-IQA: GROUNDING MULTIMODAL LANGUAGE MODEL FOR IMAGE QUALITY ASSESSMENT

**Zheng Chen**[1], **Xun Zhang**[1], **Wenbo Li**[2], **Renjing Pei**[3], **Fenglong Song**[3],
**Xiongkuo Min**[1], **Xiaohong Liu**[1], **Xin Yuan**[4], **Yong Guo**[5], **Yulun Zhang**[1*]
[1]Shanghai Jiao Tong University, [2]Joy Future Academy,
[3]Huawei Noah's Ark Lab, [4]Westlake University, [5]Huawei

## ABSTRACT

The development of multimodal large language models (MLLMs) enables the evaluation of image quality through natural language descriptions. This advancement allows for more detailed assessments. However, these MLLM-based IQA methods primarily rely on general contextual descriptions, sometimes limiting fine-grained quality assessment. To address this limitation, we introduce a new image quality assessment (IQA) task paradigm, **grounding-IQA**. This paradigm integrates multimodal referring and grounding with IQA to realize more fine-grained quality perception, thereby extending existing IQA. Specifically, grounding-IQA comprises two subtasks: grounding-IQA-description (GIQA-DES) and visual question answering (GIQA-VQA). GIQA-DES involves detailed descriptions with precise locations (e.g., bounding boxes), while GIQA-VQA focuses on quality QA for local regions. To realize grounding-IQA, we construct a corresponding dataset, GIQA-160K, through our proposed automated annotation pipeline. Furthermore, we develop a well-designed benchmark, GIQA-Bench. The benchmark evaluates the grounding-IQA performance from three perspectives: description quality, VQA accuracy, and grounding precision. Experiments demonstrate that our proposed method facilitates the more fine-grained IQA application. Code: https://github.com/zhengchen1999/Grounding-IQA.

## 1 INTRODUCTION

Image quality assessment (IQA) seeks to evaluate image quality in alignment with human perception. As a fundamental task in low-level vision, IQA is critical across multiple fields, *e.g.*, image processing (Zhang et al., 2018; Lin et al., 2019), media transmission (Ying et al., 2020), and generative artificial intelligence (Li et al., 2023). However, this task is challenging since the human visual system is inherently subjective and complex to model (Wang et al., 2004). To enhance evaluation precision, substantial research efforts continue to be dedicated to this area (Mittal et al., 2012a; Ding et al., 2020; Wang et al., 2023; Wu et al., 2024b).

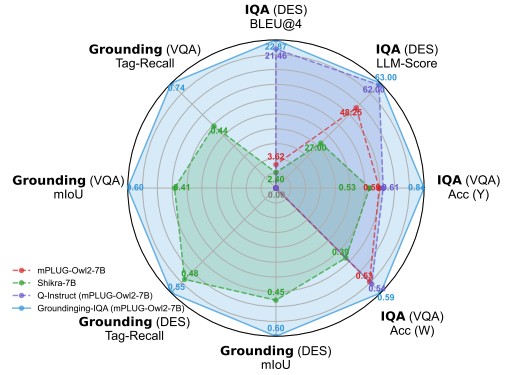

Figure 1: Performance comparisons on GIQA-Bench. Our proposed grounding-GPT effectively combines grounding and IQA.

Traditional IQA methods employ handcrafted metrics to estimate quality scores (Wang et al., 2004; Mittal et al., 2012b). With advancements in deep neural networks, learning specific priors from large datasets enables more accurate score predictions (Kang et al., 2014; Bosse et al., 2017; Jinjin et al., 2020; Ke et al., 2021). Nevertheless, score-based IQA methods face challenges in complex scenarios. In such cases, image quality is influenced by multiple factors that a single score cannot effectively express (You et al., 2024c). Recently, the emergence of multimodal large language models (MLLMs) (Liu et al., 2023; Peng et al., 2024; Ye et al., 2024) with strong visual and linguistic perception capabilities provides an alternative to score-based IQA (Wu et al., 2024a;e). These models achieve more detailed and accurate image assessments through description and reasoning.

---

*Corresponding author: Yulun Zhang, yulun100@gmail.com

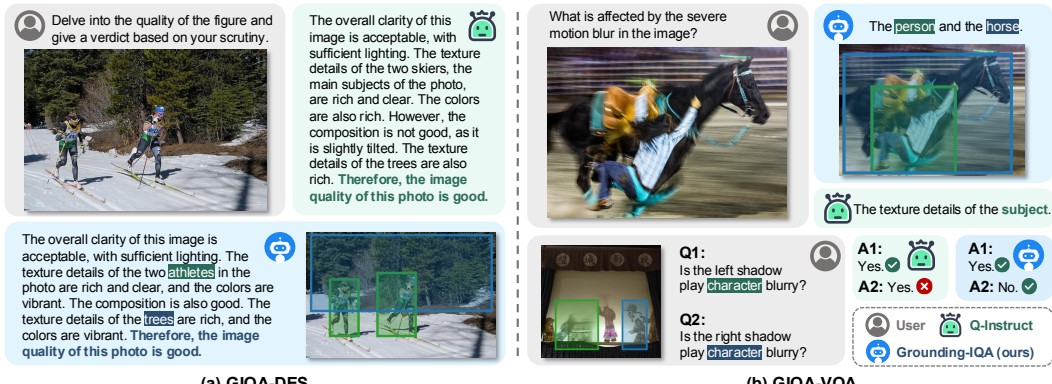

Figure 2: Grounding-IQA combines referring and grounding with IQA. (a) GIQA-DES: Quality description include precise locations (*i.e.*, bounding boxes). (b) GIQA-VQA: The question (referring, bottom instance) or answer (grounding, top instance) contains locations.

However, current MLLM-based IQA methods (Wu et al., 2024b; You et al., 2024b) primarily rely on general contextual descriptions, which sometimes limits fine-grained quality assessments. For instance, in Fig. 2a, the existing method (*i.e.*, Q-Instruct (Wu et al., 2024b)) describes the objects/areas affecting image quality through language, but cannot provide precise location information. Moreover, in Fig. 2b, for local perception, the language referring may not accurately pinpoint the target, leading to bias. These limitations restrict the application of MLLMs in comprehensive low-level perception and understanding, especially for fine-grained cases.

To address these challenges and unleash the potential of MLLMs in fine-grained image quality understanding, we introduce grounding-IQA. This is a novel IQA task paradigm that integrates multimodal referring (*position in*) and grounding (*position out*) (Mao et al., 2016; Chen et al., 2023; Peng et al., 2024) with image quality assessment. This new paradigm can serve as an extension and enhancement to existing IQA methods. Specifically, we categorize grounding-IQA into two subtasks: **(1) Grounding-IQA-Description (GIQA-DES).** As illustrated in Fig. 2a, this task requires generating descriptive assessments of image quality while providing precise locations (*i.e.*, bounding boxes) for important objects/regions impacting quality. **(2) Grounding-IQA-Visual Question Answering (GIQA-VQA).** As shown in Fig. 2b, this task involves QA about low-level attributes of images, especially regarding local objects. It includes addressing questions with specific coordinates (*referring*) or providing answers with precise positions (*grounding*).

Since existing datasets can not realize grounding-IQA well (Liu et al., 2023; You et al., 2024a; Wu et al., 2024b), we construct a new dataset, **GIQA-160K**, based on the proposed paradigm. This dataset can enhance the grounding-IQA capabilities of current MLLMs. The dataset comprises 160K instruction-tuning data with 40K images from diverse domains. Specifically, the dataset corresponds to two sub-tasks: GIQA-DES includes 60K corresponding data, and GIQA-VQA contains 100K related data. To construct the corresponding dataset, we design an **automated annotation pipeline**. The automated pipeline generates the GIQA-160K through the public IQA dataset (Wu et al., 2024b; You et al., 2024b) (with the human-annotated description). **(1) For GIQA-DES.** The task includes detailed descriptions with coordinates. We generate the data through advanced vision (Liu et al., 2024c) and language (Dubey et al., 2024) models. Through these models, we extract and filter objects and corresponding coordinates from existing descriptions and images. Meanwhile, coordinates are expressed in natural language and attached to text. This avoids extra specialized tokens and ensures data compatibility. **(2) For GIQA-VQA.** Inspired by previous work (Wu et al., 2024b; You et al., 2024a; Li et al., 2024), we construct the required data from the detailed descriptions in GIQA-DES via the LLM. We use specific QA templates (*i.e.*, "Yes/No", abbreviated as Y; "What/How/Why", abbreviated as W) and emphasize location-specific objects to generate appropriate data. The coordinates are also combined with the generated QA.

Fine-tuning on the GIQA-160K dataset enables existing pre-trained MLLMs to achieve impressive grounding-IQA capabilities. As shown in Fig. 2, the fine-tuned model can ground key objects affecting image quality, and perform more fine-grained assessments based on reference coordinates. Moreover, to comprehensively evaluate the model performance on the grounding-IQA task, we propose a well-designed benchmark, **GIQA-Bench**. This benchmark includes 100 varying types and

quality images, corresponding to 100 GIQA-Des and 150 GIQA-VQA test samples. Each sample is annotated over multiple rounds by at least three experts. We quantitatively assess grounding-IQA performance in three aspects: **(1)** assessment description quality (*i.e.*, BLEU@4, LLM-Score); **(2)** VQA accuracy (*i.e.*, Accuracy); and **(3)** grounding precision (*i.e.*, mIoU, Tag-Recall). We test recent MLLMs, with results shown in Fig. 1. Observations indicate significant improvement in grounding-IQA after fine-tuning with GIQA-160K. Overall, our contributions are threefold:

- We introduce multimodal referring and grounding into IQA, establishing a new IQA paradigm, grounding-IQA, for fine-grained quality perception and assessment.
- We construct a high-quality dataset, GIQA-160K, with an automated annotation pipeline. The dataset is versatile and suitable for fine-tuning existing MLLMs.
- We propose a high-quality benchmark, GIQA-Bench, to comprehensively evaluate the model performance on grounding-IQA from three aspects.

## 2 RELATED WORK

### 2.1 IMAGE QUALITY ASSESSMENT

**Score-based Methods.** Most current IQA methods are score-based. Early IQA approaches compute scores through handcrafted image data metrics (Wang et al., 2004; Moorthy & Bovik, 2011; Mittal et al., 2012a). However, these methods show a gap in quality perception compared to human judgment and are unsuitable for complex scenarios. With the development of the neural network, learning-based IQA methods have gradually become mainstream (Yang et al., 2022; Chen et al., 2024a; Shin et al., 2024). These methods leverage data-driven training to achieve more accurate quality assessments. For example, LPIPS (Zhang et al., 2018) applies the convolutional neural network to compute scores. Moreover, meta-learning (Zhu et al., 2020), multimodal models (Wang et al., 2023; Zhang et al., 2023c), and graph neural networks (Sun et al., 2022) have been adopted to further improve IQA. However, score-based IQA methods face limitations in complex scenarios. The simple score cannot effectively represent the multiple aspects affecting image quality.

**MLLM-based Methods.** Multimodal large language models (MLLMs) exhibit remarkable multimodal (language/vision) understanding by integrating visual modules into LLMs (Liu et al., 2023; Zhang et al., 2023a; Jiang et al., 2024). MLLMs achieve outstanding performance in various multimodal tasks, including visual question answering and image captioning. Recently, several studies have also demonstrated the potential of MLLMs in low-level visual perception and assessment (Wu et al., 2024b; You et al., 2024b;b; Wu et al., 2024d; Chen et al., 2024b). For instance, Q-Instruct (Wu et al., 2024b) constructs a multimodal dataset to enhance. Q-Align (Wu et al., 2024c) guides MLLMs in scoring by defining discrete text-based levels. DepictQA (You et al., 2024c) enables quality comparison and reasoning based on reference images. These approaches advance the application of MLLMs in IQA, achieving more accurate assessments. Nevertheless, these models primarily rely on contextual descriptions, and face limitations in fine-grained applications, *e.g.*, local perception.

### 2.2 MULTIMODAL REFERRING AND GROUNDING

Multimodal spatial perception involves referring and grounding. **Referring** requires the model to understand the specific region based on position input, *e.g.*, region-level captioning (Krahmer & Van Deemter, 2012; Zellers et al., 2019). **Grounding**, on the other hand, involves the model describing the region by outputting position, *e.g.*, referring expression comprehension (Kazemzadeh et al., 2014; Luo & Shakhnarovich, 2017). Currently, MLLMs perform impressively in spatial perception, further advancing these tasks. Some methods focus on grounding, achieving complex reasoning (Lai et al., 2024) or multi-object (Ren et al., 2024) segmentation. Meanwhile, other approaches, *e.g.*, GPT4RoI (Zhang et al., 2023b), emphasize understanding specific regions (referring). Furthermore, some works unify referring and grounding (Chen et al., 2023; Li et al., 2024; Rasheed et al., 2024; Peng et al., 2024; You et al., 2024a). Additionally, in IQA, Q-Ground (Chen et al., 2024b) achieves degradation region grounding but lacks referring capabilities. In contrast, our Grounding-IQA integrates multimodal referring and grounding with IQA to enhance quality perception.

## 3 METHOD

In this section, we introduce the newly defined IQA paradigm, grounding-IQA. The content includes: **(1)** definition of paradigm and two subtasks, Sec. 3.1; **(2)** data construction pipeline, Sec. 3.2; **(3)** details of GIQA-160K, Sec. 3.3; **(4)** benchmark for grounding-IQA, Sec. 3.4.

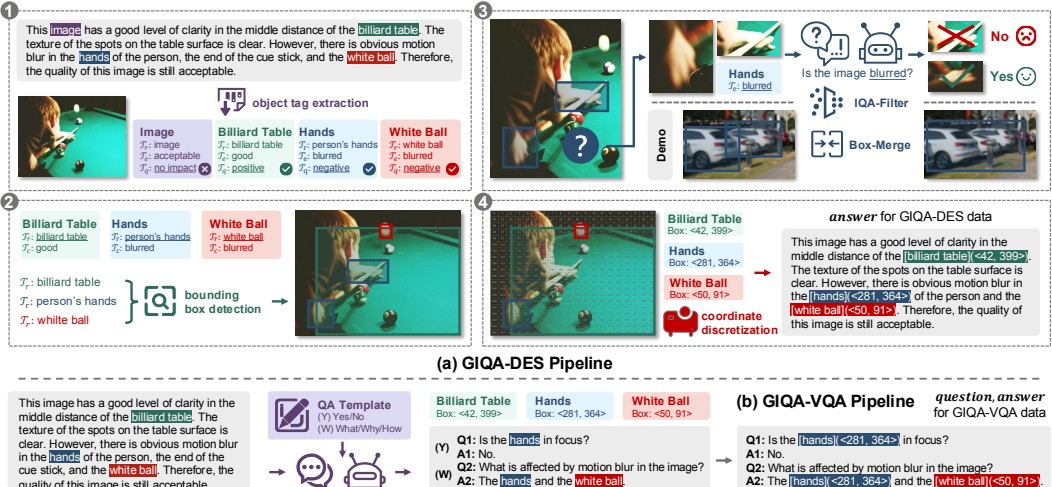

Figure 3: The illustration of the automated annotation pipeline. (a) GIQA-DES Pipeline: Constructs the **answer** from the given image and description via a four-stage process, while the **question** comes from a predefined question pool. (b) GIQA-VQA Pipeline: Generates the corresponding QA data utilizing descriptions from GIQA-DES and the LLM (Llama3 (Dubey et al., 2024)).

## 3.1 GROUNDING-IQA

As analyzed above, existing MLLM-based IQA methods leverage descriptions to enable more accurate and detailed quality assessments. However, these methods remain limited in performing fine-grained evaluations, as in Fig. 2. Inspired by work on multimodal referring and grounding, we believe that spatial perception is key to achieving more fine-grained assessments. Therefore, to further unlock the potential of MLLMs, we introduce a new IQA paradigm, grounding-IQA. This paradigm combines referring and grounding with IQA to enable more precise and flexible quality assessments. Specifically, grounding-IQA should include the two sub-tasks/capabilities: grounding-IQA-description (GIQA-DES) and grounding-IQA-visual question answering (GIQA-VQA).

**GIQA-DES.** The task requires the model to provide a detailed description of image quality. Additionally, it needs accurate location information (*e.g.*, bounding box) for key objects/regions that impact image quality, as shown in Fig. 5a. This corresponds to the fact that humans consider not only the overall quality (*e.g.*, image clarity) but also the quality of specific objects or locations when assessing image quality. Meanwhile, accurate location information also enables targeted information for downstream tasks (*e.g.*, image editing). This task is similar to grounded image captioning (Zhou et al., 2020), but places greater emphasis on low-level attributes. While some MLLMs (Chen et al., 2023; Peng et al., 2024; Li et al., 2024) perform well in grounded image captioning, they still struggle with quality perception. We demonstrated it in Sec. 4.3.

**GIQA-VQA.** The second task focuses on the question-answering ability in low-level perception, particularly for local objects. Corresponding to multimodal referring and grounding, this task can be divided into two scenarios. **Referring:** querying low-level attributes in the specified region (*input position*), as shown in Fig. 5b. **Grounding:** providing answers that include specific locations (*output position*) based on the question, as depicted in Fig. 5b. These two scenarios are related to region captioning (Zhou et al., 2020) and phrase grounding (Zhou et al., 2020), respectively. However, like GIQA-DES, GIQA-VQA involves quality perception, which is challenging for current MLLMs.

## 3.2 AUTOMATED ANNOTATION PIPELINE

Data is essential for achieving Grounding-IQA. Therefore, we construct an automated annotation pipeline to generate data (*i.e.*, GIQA-160K). This pipeline leverages public IQA datasets (Wu et al., 2024b; You et al., 2024b) that contain human-annotated descriptions. Following previous schemes (Liu et al., 2023; Ye et al., 2024), the data format is {*image*, *question*, *answer*}. The *image* is the evaluation target. Depending on the sub-task, the *question* and *answer* fields may include precise coordinates (*i.e.*, **bounding box**), in addition to text. The illustration of the whole pipeline is in Fig. 3. Besides, more details are provided in the supplementary material.

**For GIQA-DES.** In this task, the *question* is relatively fixed, as in Fig. 5a. For each data point, the *question* is randomly selected from the question pool with 15 similar questions. For the *answer*, it is a detailed description with coordinates. We construct it via a four-stage process from existing images and associated description, as illustrated in Fig. 3: **(1) Stage-1:** object tag extraction; **(2) Stage-2:** bounding box detection; **(3) Stage-3:** box refinement (filter and merge); and **(4) Stage-4**: transformation and fusion. Each stage is detailed below.

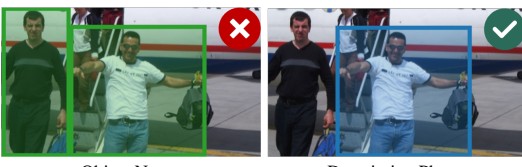

Figure 4: Utilizing the description phrase $\mathcal{T}_r$ ("the man wearing a white t-shirt") yields more accurate detection than applying object name ("man").

***Stage-1: Object Tag Extraction.*** Firstly, we apply the advanced LLM, *i.e.*, Llama3 (Dubey et al., 2024), to extract key objects (*e.g.*, "billiard table" in Fig. 3a) from the given descriptions. Each object is assigned a three-tuple form tag: $\{\mathcal{T}_r, \mathcal{T}_q, \mathcal{T}_e\}$. The $\mathcal{T}_r$ is the object description phrase (sometimes same as name); $\mathcal{T}_q$ denotes the quality of object (*e.g.*, "clear"); $\mathcal{T}_e$ represents the object effect on image quality (*i.e.*, "no impact", "positive", or "negative"). All tag items are inferred from the description, with $\mathcal{T}_r$ and $\mathcal{T}_q$ used in later stages. The $\mathcal{T}_e$ item enables us to filter out non-critical objects (*e.g.*, "image", which refers to the whole). This explicit effect classification, similar to chain-of-thought (CoT), can reduce hallucinations.

***Stage-2: Bounding Box Detection.*** Then, we detect bounding boxes for the extracted objects from the image. To accomplish this, we utilize the state-of-the-art object detection model, Grounding DINO (Liu et al., 2024c). Since multiple same-category objects may appear in one image, we utilize the $\mathcal{T}r$ generated **Stage-1** rather than the object name for detection. For instance, in Fig. 4, the object name is "man", and $\mathcal{T}r$ is "the man wearing a white t-shirt". Leveraging "man" detects two objects (left case), while using $\mathcal{T}r$ can achieve the more precise result (right case).

***Stage-3: Box Refinement.*** Although **Stage-2** adopts $\mathcal{T}r$ to limit the detection range, multiple boxes may still exist. In some cases, multiple boxes may contain the wrong target. Through observations, most detection errors arise from the detection model inability to distinguish objects of same class with different quality. For instance, in Fig. 3a, for "hands", the key (reduce image quality) is the blurry one, and the other is irrelevant. To address this problem, we design the IQA-Filter algorithm (Alg. 1). We use the MLLM-based IQA method, Q-Instruct, to verify detected bounding boxes by inputting each box patch and asking: "Is the image quality is <$\mathcal{T}_q$>?", with $\mathcal{T}_q$ from **Stage-1**. We check all boxes in single-object-multiple-targets, and remove those with a "No" response.

Furthermore, in some cases, multiple small or overlapping targets correspond to the same object. While these detections are accurate, an excess of targets may increase the learning difficulty for MLLMs. To address this issue, we propose the Box-Merge algorithm (Alg. 1). We merge boxes that satisfy the normalized area threshold $T_a$ (set to 0.256), and the overlap threshold $T_o$ (set to 95%).

***Stage-4: Transformation and Fusion.*** Finally, we integrate the extracted and filtered boxes into the original descriptions to construct the *answer*. To avoid introducing extra specialized tokens for box representation, we treat box coordinates as regular text tokens, attaching them to the text in the interleaved format: "[object/region](bounding box)".

Moreover, bounding boxes are typically represented by normalized corner coordinates: $\langle x_1, y_1, x_2, y_2 \rangle$. When the coordinate values are rounded to two decimal places (*e.g.*, $\langle 0.01, 0.02, 0.03, 0.04 \rangle$), representing box requires **21** tokens. Inspired by previous work (You et al., 2024a; Peng et al., 2024), we discretize the coordinates for simplicity. We divide the image into $n \times m$ grids and numbering grids from top-left to bottom-right: $\{0,1,\ldots,nm-1\}$. Patch numbers then represent the top-left and bottom-right coordinates of the box:

$$\text{idx}_l = y_1 \cdot m \cdot n + x_1 \cdot n, \quad \text{idx}_r = y_2 \cdot m \cdot n + x_2 \cdot n, \tag{1}$$

where $\text{idx}_l$ and $\text{idx}_r$ denotes the coordinates. The box can be represented as $\langle \text{idx}_l, \text{idx}_r \rangle$. Accordingly, we remap the discrete coordinates back to a continuous format using the centre coordinates:

$$x_1' = (\text{idx}_l \% n + 0.5)/n, \ y_1' = (\text{idx}_l/n + 0.5)/m,$$
$$x_2' = (\text{idx}_r \% n + 0.5)/n, \ y_2' = (\text{idx}_r/n + 0.5)/m, \tag{2}$$

where new coordinates is $\langle x_1', y_1', x_2', y_2' \rangle$. Though the discretization reduces coordinate precision, it effectively simplifies the representation. In our dataset, we set $n=m=20$, requiring at most **9** tokens.

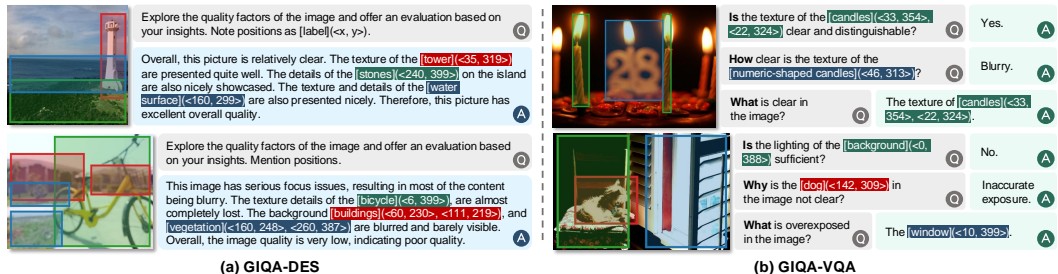

Figure 5: Some instances from the GIQA-160K, involving subtasks: GIQA-DES and GIQA-VQA.

Finally, the **answer** is a natural language description with precise coordinates, as shown in Fig. 3a.

**For GIQA-VQA.** The task requires that the **question** or **answer** relate to low-level attributes and include explicit spatial information (*i.e.*, bounding boxes). Inspired by previous work (Wu et al., 2024b; You et al., 2024a; Li et al., 2024), we apply the LLM (*i.e.*, Llama3 (Dubey et al., 2024)) to generate the corresponding QA pairs from the descriptions in GIQA-DES (depicted in Fig. 3b). We use specific templates to generate diverse QA. Details are as follows:

*(1) Binary Questions ("Yes/No"):* Answers are limited to "Yes" or "No". The "Yes" answer corresponds to questions inferred directly from the description. Conversely, quality questions that cannot be inferred are answered "No".

---

**Algorithm 1** IQA-Filter & Box-Merge

1: **Input:** target image $I$, object bounding boxes $\mathcal{B}$, object quality $\mathcal{T}_q$, area threshold $T_a$, overlap threshold $T_o$
2: **Output:** the refined bounding boxes $\mathcal{R}$
3: **Init:** $\mathcal{R} \leftarrow \emptyset$
   ▷ IQA-Filter: filter boxes by quality query
4: **for** $b \in \mathcal{B}$ **do**
5:    $p \leftarrow \text{patch}(I, b)$; $q \leftarrow$ "Is the image quality $<\mathcal{T}_q>$?"
6:    **if** Q-Instruct$(p, q) = $ 'Yes" **then**
7:      $\mathcal{R} \leftarrow \mathcal{R} \cup \{b\}$
8:    **end if**
9: **end for**
   ▷ Box-Merge: merge overlapped boxes
10: **for** $i = 0; i < |\mathcal{R}|; i \leftarrow i + 1$ **do**
11:    $j \leftarrow i + 1$
12:    **while** $j < |\mathcal{R}|$ **do**
13:      **if** area$(\mathcal{R}[i]) < T_a$ **and** is-touch$(\mathcal{R}[i], \mathcal{R}[j])$ **or** coverage-ratio$(\mathcal{R}[i], \mathcal{R}[j]) > T_o$ **then**
14:        $\mathcal{R}[i] \leftarrow \text{merge}(\mathcal{R}[i], \mathcal{R}[j])$; $\mathcal{R} \leftarrow \mathcal{R} \setminus \{\mathcal{R}[j]\}$
15:      **else**
16:        $j \leftarrow j + 1$
17:      **end if**
18:    **end while**
19: **end for**
20: **return** $\mathcal{R}$

---

*(2) Open-ended Questions ("What/Why/How"):* These questions address low-level attributes or related context (*e.g.*, "What types of distortion?"); cause analysis (*e.g.*, "Why the image quality is poor?"); perceptual degree (*e.g.*, "How is clarity?"). All answers are inferred from the description and given as short phrases (*e.g.*, "Noise" and "Medium" ).

Meanwhile, we supply the LLM with the names of key objects/regions (with bounding boxes), constraining the QA to relate to relevant entities. We also use keyword detection to filter out any unrelated QA pairs. Finally, we incorporate bounding box information into the generated QA pairs, forming the corresponding **question** and **answer**.

### 3.3 GIQA-160K

We construct our grounding-IQA dataset, GIQA-160K, utilizing the automated annotation pipeline, from existing public datasets (Wu et al., 2024b; You et al., 2024b). Figure 5 shows some instances.

**Data Source.** To build our dataset, we require two types of data: diverse images and their corresponding detailed quality descriptions. Currently, two public datasets, Q-Pathway (Wu et al., 2024b) and DQ-495K (You et al., 2024b), meet our requirements. For Q-Pathway, we select in-the-wild images (KonIQ-10K (Hosu et al., 2020), SPAQ (Fang et al., 2020), LIVE-FB (Ying et al., 2020), and LIVE-itw (Ghadiyaram & Bovik, 2015)) and AI-generated images (AGIQA-3K (Li et al., 2023) and ImageRewardDB (Xu et al., 2024)), along with their professionally human-annotated texts. The total image-text pairs is 53K. For DQ-495K, 27K artificially degraded images (from KADIS-700K (Lin et al., 2020)) are paired with human-annotated descriptive texts.

**Dataset Statistic.** Utilizing the above raw data (80K image-text pairs), we construct a dataset with **167,657** instruction-tuning samples and **42,960** images. Dataset statistics are shown in Tab. 1. For GIQA-DES, we generate 66,689 detailed quality descriptions with coordinates. The GIQA-VQA contains 100,968 question-

Table 1: Statistics information of the proposed datasets. DES: GIQA-DES; VQA: GIQA-VQA.

| Dataset | Image | Total | DES | VQA (Y) | VQA (W) |
|---|---|---|---|---|---|
| GIQA-160K | 42,960 | 167,657 | 66,689 | 50,484 | 50,484 |
| GIQA-Bench | 100 | 250 | 100 | 90 | 60 |

answer pairs. For GIQA-VQA, to balance question types, we randomly filter to maintain an equal amount of "Yes/No" and "What/Which/How" questions (50,484 each). Additionally, we ensured a balanced distribution between "Yes" and "No" responses, with 25,242 samples in each category.

## 3.4 GIQA-BENCH

We construct a high-quality benchmark, **GIQA-Bench**, to evaluate the model grounding-IQA performance, detailing its data statistics and evaluation criteria.

**Bench Statistic.** The GIQA-Bench includes 100 images of various types and quality, which are not included in GIQA-160K. We create 100 GIQA-DES and 150 GIQA-VQA test samples based on these images. Among the 150 GIQA-VQA data, 90 are of the "Yes/No" questions ("Yes": 35; "No": 55), and 60 are "What/Which/How" questions ("What": 30; "Why": 18; "How": 12).

The descriptions for GIQA-DES are from Q-Pathway and adjusted, with key objects and bounding boxes manually determined. GIQA-VQA questions are generated by the annotation pipeline and further refined and answered by humans. Each sample is annotated in multiple rounds by at least three experts with relevant expertise in a controlled laboratory environment to ensure accuracy.

**Evaluation Criteria.** We evaluate the grounding-IQA capabilities from three perspectives: description quality, VQA accuracy, and grounding precision. For all metrics, higher values are better.

*(1) Description Quality.* Assess GIQA-DES performance in quality descriptions. We compare the generated description to the ground truth, excluding coordinates. We apply the image captioning metric: ***BLEU@4***. We also employ the LLM (Llama3 (Dubey et al., 2024)) to provide a score from 0 to 4 (higher is better), based on the relevance between the description and the ground truth. For clarity, the final score is scaled proportionally from 0 to 100. We denote the score as the ***LLM-Score***.

*(2) VQA Accuracy.* Evaluate GIQA-VQA performance in quality VQA. For "Yes/No" questions, accuracy is determined by matching with the word "Yes" or "No". For "What/Which/How", we use LLM to calculate accuracy. The LLM scores the model response from 0 to 4 (higher is better) based on the question and correct answer. The score is normalized to 0~1. We denote the accuracy of "Yes/No" as ***Acc (Y)***, "What/Which/How" as ***Acc (W)***, and overall accuracy as ***Acc (Total)***.

*(3) Grounding Precision.* Measure the grounding performance for both GIQA-DES and GIQA-VQA. We use category-agnostic mean Intersection over Union (***mIoU***) to evaluate box quality. We also define ***Tag-Recall*** to assess category-specific grounding capabilities. In Tag-Recall, a result is true positive only if both the IoU and object name similarity exceeds a 0.5 threshold. For fairness, the bounding box is represented by the normalized corner coordinate.

## 4 EXPERIMENTS

### 4.1 EXPERIMENTAL SETTINGS

**Implementation Details.** We conduct experiments on four pre-trained MLLM models: LLaVA-v1.5-7B (Liu et al., 2024a), LLaVA-v1.5-13B (Liu et al., 2024a), LLaVA-v1.6-7B (Liu et al., 2024b), and mPLUG-Owl2-7B (Ye et al., 2024). These models involve different versions, sizes, and architectures. The models are fine-tuned on our proposed GIQA-160K dataset using supervised fine-tuning. We evaluate their performance on grounding-IQA using the GIQA-Bench. Details about the training/testing datasets and evaluation criteria are provided in Secs. 3.3 and 3.4.

**Training Settings.** We adopt cross-entropy loss for full fine-tuning, following previous methods (Wu et al., 2024b; Liu et al., 2023; Ye et al., 2024). The optimizer is AdamW (Loshchilov et al., 2018), with $\beta_1$=0.9 and $\beta_2$=0.999. We apply the cosine decay scheduler with an initial learning rate of $2\times10^{-5}$, and a warmup ratio of 0.03. The batch size is set to 64, and the epoch is 2. Other hyper-parameters follow the default settings of each model. Experiments are implemented with PyTorch (Paszke et al., 2019) on four Nvidia A100-80G GPUs.

Table 2: Ablation study on box optimization (refinement and representation) in the automated annotation pipeline. We conduct experiments on the GIQA-DES task.

(a) Box refinement.

| Method | mIoU | Tag-Recall | BLEU@4 | LLM-Score |
|--------|------|-----------|--------|-----------|
| Baseline | N/A | N/A | 3.62 | 48.25 |
| Raw-Box | 0.5624 | 0.5045 | 20.97 | 61.00 |
| Ref-Box | **0.5851** | **0.5497** | **23.67** | **61.75** |

(b) Box representation.

| Method | mIoU | Tag-Recall | BLEU@4 | LLM-Score |
|--------|------|-----------|--------|-----------|
| Baseline | N/A | N/A | 3.62 | 48.25 |
| Norm-Coord | **0.6046** | 0.5490 | 22.03 | 61.00 |
| Disc-Coord | 0.5851 | **0.5497** | **23.67** | **61.75** |

Table 3: Ablation study on multi-task training. The baseline is the pre-trained model, mPLUG-Owl2-7B, without fine-tuning.

| Method | GIQA-DES | | GIQA-VQA | |
|--------|----------|--|----------|--|
| | Tag-Recall | LLM-Score | Tag-Recall | Acc (Total) |
| Baseline | N/A | 48.25 | N/A | 0.5633 |
| Only-DES | **0.5497** | 61.75 | 0.5577 | 0.5900 |
| Only-VQA | 0.3283 | 38.50 | 0.4872 | 0.7217 |
| GIQA-160K | 0.5474 | **63.00** | **0.7372** | **0.7417** |

Figure 6: Box area distribution of GIQA-160K (Raw and Ref) and GIQA-Bench.

## 4.2 ABLATION STUDY

We analyze method design and data properties. The training settings are detailed in Sec. 4.1. We apply mPLUG-Owl2-7B (Ye et al., 2024) as the baseline in all experiments (except in Tab. 4).

**Box Optimization.** We evaluate box optimization in the annotation pipeline, including the box refinement (IQA filter and box merge) and the coordinate representation. We compare the models trained on GIQA-DES with (Ref-Box) and without refinement (Raw-Box) in Tab. 2a. The refinement enhances the fine-tuning effect. We also visualize box area distribution in Fig. 6. Refinement reduces the difference between automatically annotated GIQA-160K and human-annotated GIQA-Bench. Besides, more analyses are provided in the supplementary material.

Meanwhile, we compare discrete (Disc-Coord) and normalized continuous (Norm-Coord) box representations in Tab. 2b. Results indicate that Disc-Coord enhances description quality (BLEU@4 and LLM-Score) and grounding accuracy (Tag-Recall), compared with Norm-Coord.

**Multi-Task Training.** We conduct an ablation on multi-task (GIQA-DES and GIQA-VQA) joint training. The results are listed in Tab. 3. We observe that only GIQA-DES (Only-DES) can improve the quality assessment and grounding. GIQA-VQA improves VQA accuracy but exhibits limited grounding ability, likely due to reduced contextual information compared to GIQA-DES. Moreover, multi-task training (GIQA-160K) enhances performance on both GIQA-DES and GIQA-VQA. It demonstrates the importance of data diversity.

Table 4: Ablation study on different baselines.

| Method | SFT | GIQA-DES | | GIQA-VQA | |
|--------|-----|----------|--|----------|--|
| | | Tag-Recall | LLM-Score | Tag-Recall | Acc (Total) |
| LLaVA-1.5-7B | | N/A | 47.00 | N/A | 0.4733 |
| | ✓ | **0.5283** | **60.00** | **0.5961** | **0.6850** |
| LLaVA-1.5-13B | | N/A | 49.00 | N/A | 0.4433 |
| | ✓ | **0.5548** | **60.50** | **0.7564** | **0.6950** |
| LLaVA-1.6-7B | | N/A | 50.50 | N/A | 0.5067 |
| | ✓ | **0.5981** | **60.00** | **0.6538** | **0.7250** |
| mPLUG-Owl-2-7B | | N/A | 48.25 | N/A | 0.5633 |
| | ✓ | **0.5474** | **63.00** | **0.7372** | **0.7417** |

**Data Compatibility.** We fine-tune various baselines using the proposed GIQA-160K. The results are provided in Tab. 4. The results indicate that our proposed dataset is compatible with various MLLMs, effectively enhancing the grounding-IQA ability of the model. Furthermore, we provide more detailed comparisons with more methods in Sec. 4.3.

## 4.3 RESULTS ON GIQA-BENCH

In GIQA-Bench, we compare four groups of MLLMs with different functionalities, *i.e.*, **(1)** General models (General): LLaVA-v1.5-7B (Liu et al., 2024a), LLaVA-v1.5-13B (Liu et al., 2024a), LLaVA-v1.6-7B (Liu et al., 2024b), and mPLUG-Owl2-7B (Ye et al., 2024); **(2)** Multimodal referring and grounding models (Ground): Shikra-7B (Chen et al., 2023), Kosmos-2-1.6B (Peng et al., 2024), Ferret-7B (You et al., 2024a), and GroundingGPT-7B (Li et al., 2024); **(3)** IQA models (IQA): DepictQA-Wild-7B (You et al., 2024b) and Q-Instruct (Wu et al., 2024b) (fine-tuned three base models); and **(4)** Our methods (Ours): Four general models fine-tuned on GIQA-160K. The detailed **test settings** and **analyses** are provided in the supplementary material.

| Group | Method | GIQA-DES | | | | GIQA-VQA | | | | |
|---|---|---|---|---|---|---|---|---|---|---|
| | | mIoU | Tag-Recall | BLEU@4 | LLM-Score | mIoU | Tag-Recall | Acc (Y) | Acc (W) | Acc (Total) |
| General | LLaVA-v1.5-7B | N/A | N/A | 2.82 | 47.00 | N/A | N/A | 0.4444 | 0.5167 | 0.4733 |
| | LLaVA-v1.5-13B | N/A | N/A | 3.00 | 49.00 | N/A | N/A | 0.3888 | 0.5250 | 0.4433 |
| | LLaVA-v1.6-7B | N/A | N/A | 3.04 | 50.50 | N/A | N/A | 0.4889 | 0.5333 | 0.5067 |
| | mPLUG-Owl2-7B | N/A | N/A | 3.62 | 48.25 | N/A | N/A | 0.5889 | 0.5250 | 0.5633 |
| Ground | Shikra-7B | 0.4506 | 0.4768 | 0.40 | 27.00 | 0.4126 | 0.4359 | 0.5333 | 0.3917 | 0.4767 |
| | Kosmos-2-1.6B | 0.4946 | 0.3448 | 2.63 | 39.25 | 0.4982 | 0.4103 | 0.3889 | 0.4750 | 0.4233 |
| | Ferret-7B | 0.6458 | 0.6778 | 3.16 | 43.75 | 0.5393 | 0.5769 | 0.4111 | 0.4875 | 0.4417 |
| | GroundingGPT-7B | 0.4967 | 0.5391 | 1.99 | 32.50 | 0.3845 | 0.5321 | 0.5444 | 0.5250 | 0.5367 |
| IQA | DepictQA-Wild-7B | N/A | N/A | 3.34 | 56.50 | N/A | N/A | 0.4333 | 0.5458 | 0.4783 |
| | Q-Instruct (LLaVA-v1.5-7B) | N/A | N/A | 22.69 | 58.25 | N/A | N/A | 0.6444 | 0.5375 | 0.6017 |
| | Q-Instruct (LLaVA-v1.5-13B) | N/A | N/A | 19.01 | 57.25 | N/A | N/A | 0.6222 | 0.5417 | 0.5900 |
| | Q-Instruct (mPLUG-Owl2-7B) | N/A | N/A | 21.46 | 62.00 | N/A | N/A | 0.6111 | 0.5375 | 0.5817 |
| Ours | Grounding-IQA (LLaVA-v1.5-7B) | 0.5763 | 0.5283 | 19.02 | 60.00 | 0.5180 | 0.5961 | 0.7777 | 0.5458 | 0.6850 |
| | Grounding-IQA (LLaVA-v1.5-13B) | 0.6302 | 0.5548 | 20.24 | 60.50 | 0.6830 | 0.7564 | 0.7889 | 0.5542 | 0.6950 |
| | Grounding-IQA (LLaVA-v1.6-7B) | 0.6583 | 0.5981 | 19.17 | 60.00 | 0.5459 | 0.6538 | 0.8333 | 0.5625 | 0.7250 |
| | Grounding-IQA (mPLUG-Owl2-7B) | 0.5955 | 0.5474 | 22.87 | 63.00 | 0.6031 | 0.7372 | 0.8444 | 0.5875 | 0.7417 |

Table 5: Quantitative results on GIQA-Bench. Best and second-best results are colored red and blue.

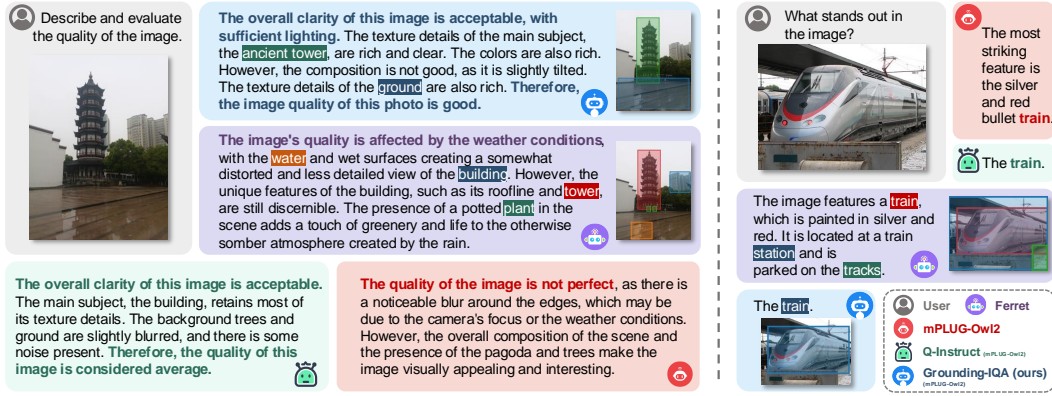

Figure 7: Visual comparisons on GIQA-Bench. Our proposed grounding-IQA (**blue module**) enables more fine-grained quality descriptions (left instance) and QA (right instance).

**Quantitative Results.** We evaluate all models on GIQA-DES and GIQA-VQA from two aspects: quality assessment and grounding ability, as in Tab. 5. General models perform poorly on both tasks, while task-specific models are more effective in their respective domains. Specifically, grounding MLLMs excel in grounding tasks but underperform on quality-related objects/areas (GIQA-VQA, Tag-Recall). Conversely, IQA models achieve high description quality (GIQA-DES, LLM-Score), but exhibit low accuracy in GIQA-VQA. In contrast, our method outperforms existing MLLMs.

Moreover, to further demonstrate the performance and generalization ability of our approach, we conduct extensive experiments and evaluations in the supplementary material, including: **(1)** traditional score-based IQA tasks; **(2)** the user study on GIQA-Bench, and **(3)** the application of grounding-IQA to downstream tasks. Our method also achieves impressive performance.

**Qualitative Results.** We provide some visual comparisons in Fig. 7. For GIQA-DES (left instance), the quality descriptions generated by general (mPLUG-Owl2-7B (Ye et al., 2024)) and grounding (Ferret (You et al., 2024a)) MLLMs are unsatisfactory. In contrast, our method describes image quality more properly with coordinates of key objects affecting the quality. Furthermore, in the GIQA-VQA task (right instance), our method produces more accurate responses to image quality VQA involving spatial perception. More results are provided in the supplementary material.

## 5 CONCLUSION

In this paper, we introduce a new IQA task paradigm called Grounding-IQA for fine-grained quality assessments. The grounding-IQA combines multimodal referring and grounding with IQA, and comprises two subtasks: GIQA-DES and GIQA-VQA. Under the task paradigm, we construct a corresponding dataset, GIQA-160K, by an automated annotation pipeline. Meanwhile, we develop a benchmark, GIQA-Bench, to evaluate the grounding-IQA. Experiments indicate that our proposed task, dataset, and benchmark facilitate more fine-grained IQA applications.

## ACKNOWLEDGMENTS

This work is supported by the National Natural Science Foundation of China (62501386, 625B2116, U2541205, 62271414, 62572317), CCF-Tencent Rhino-Bird Open Research Fund, National Key R&D Program of China (2024YFF0505603), "Pioneer" and "Leading Goose" R&D Program of Zhejiang (Grant 2024SDXHDX0006, 2024C03182), the Key Project of Westlake Institute for Optoelectronics (grant number 2023GD007), and the 2023 International Sci-tech Cooperation Projects under the purview of the "Innovation Yongjiang 2035" Key R&D Program (grant number 2024Z126).

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
