# GROUNDING-IQA: GROUNDING MULTIMODAL LANGUAGE MODEL FOR IMAGE QUALITY ASSESSMENT
## *Supplementary Material*

**Zheng Chen**[1], **Xun Zhang**[1], **Wenbo Li**[2], **Renjing Pei**[3], **Fenglong Song**[3],
**Xiongkuo Min**[1], **Xiaohong Liu**[1], **Xin Yuan**[4], **Yong Guo**[5], **Yulun Zhang**[1*]
[1]Shanghai Jiao Tong University, [2]Joy Future Academy,
[3]Huawei Noah's Ark Lab, [4]Westlake University, [5]Huawei

## OVERVIEW

In the supplementary materials, we provide more results and analyses, including:

- Sec. 1: Evaluation on score-based IQA tasks.
- Sec. 2: User Study on GIQA-Bench.
- Sec. 3: Evaluation on the downstream task: image super-resolution (SR).
- Sec. 4: Automated annotation pipeline details.
- Sec. 5: Metrics implementation and analysis.
- Sec. 6: More dataset statistics.
- Sec. 7: More details of GIQA-Bench.
- Sec. 8: Limitations and future work.
- Sec. 9: Materials for the rebuttal, including various visualizations.

## 1 EVALUATION ON SCORE-BASED IQA TASK

We evaluate our model on score-based IQA tasks to further demonstrate its IQA capabilities.

### 1.1 IMPLEMENTATION

We first introduce the strategy of applying MLLM-based methods to score-based IQA tasks. We apply the softmax-based strategy proposed in Q-Bench (Wu et al., 2024a) to extend the proposed Grounding-IQA. Specifically, we employ a conversational template for MLLM:

*#User: Rate the quality of the image [IMAGE].*
*#Assistant: The quality of the image is [SCORE_TOKEN]*

Based on the conversational template, the IQA score is then calculated as:

$$\text{IQA-Score} = e^{\text{good}}/(e^{\text{good}} + e^{\text{poor}}), \tag{1}$$

where $e^{\text{good}}$ and $e^{\text{poor}}$ are the predicted logits of the Multimodal Language Model at the [SCORE_TOKEN] position for the terms "good" and "poor".

### 1.2 RESULTS

We compare our Grounding-IQA with recent score-based IQA methods (*i.e.*, HyperIQA (Su et al., 2020), MUSIQ (Ke et al., 2021), CLIP-IQA+ (Wang et al., 2023), and Q-Align (Wu et al., 2024c)), and MLLM-based methods (*i.e.*., mPLUG-Owl2 (Ye et al., 2024), DepictQA-Wild (You et al., 2024b), and Q-Instruct (Wu et al., 2024b)) on score-based IQA benchmarks (*i.e.*, KonIQ (Hosu et al., 2020), KADID (Lin et al., 2019), and LIVE Challenge).

**For score-based IQA methods.** Following the previous setting (Ke et al., 2021; Wang et al., 2023), We train all score-based methods on a single dataset (*i.e.*, KonIQ) and evaluate them on all datasets.

**For MLLM-based IQA methods.** For Q-Instruct (Wu et al., 2024b) and our proposed Grounding-IQA, we use versions fine-tuned on mPLUG-Owl2. All MLLM-based models are trained exclusively

---

*Corresponding author: Yulun Zhang, yulun100@gmail.com

Table 1: Results on score-based IQA tasks. **Score-based methods** (top section) are trained on KonIQ, while **MLLM-based methods** (bottom section) are trained on their respective multimodal datasets without using MOS values. **Fairness**: All models (score-based or MLLM-based) are not trained on KADID or LIVE Challenge. The best and second-best results are colored red and blue.

| Method | KonIQ | | KADID | | LIVE Challenge | |
|---|---|---|---|---|---|---|
| | SRCC | PLCC | SRCC | PLCC | SRCC | PLCC |
| HyperIQA (Su et al., 2020) | 0.9060 | 0.9170 | 0.4680 | 0.5060 | 0.7490 | 0.7720 |
| MUSIQ (Ke et al., 2021) | 0.9290 | 0.9240 | 0.5560 | 0.5750 | 0.8300 | 0.7890 |
| CLIP-IQA+ (Wang et al., 2023) | 0.8950 | 0.9090 | 0.6540 | 0.6530 | 0.8050 | 0.8320 |
| Q-Align (Wu et al., 2024c) | 0.9400 | 0.9410 | 0.6840 | 0.6740 | 0.8600 | 0.8530 |
| mPLUG-Owl2-7B (Ye et al., 2024) | 0.4220 | 0.4493 | 0.5410 | 0.5460 | 0.4980 | 0.5188 |
| DepictQA-Wild (You et al., 2024b) | 0.5333 | 0.5107 | 0.7385 | 0.6816 | 0.5734 | 0.5594 |
| Q-Instruct (Wu et al., 2024b) | 0.8990 | 0.9160 | 0.6980 | 0.6760 | 0.8192 | 0.7981 |
| Grounding-IQA (ours) | 0.9342 | 0.9282 | 0.7715 | 0.7648 | 0.8781 | 0.8652 |

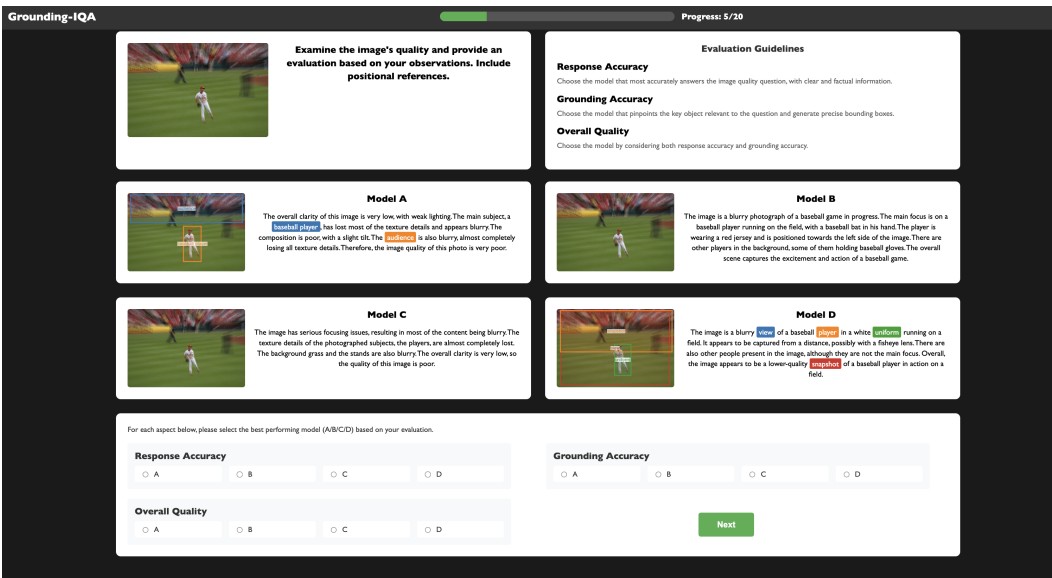

Figure 1: Interface of the anonymous website used for the user study.

on their respective image-text datasets without using any MOS values during training. We employ the softmax-based strategy to adapt them for score-based IQA tasks.

**Fairness considerations.** Since the training data for all MLLM-based methods do not include KADID or LIVE Challenge data, we ensure fairness by training score-based IQA methods on KonIQ. The other two datasets, KADID and LIVE Challenge, serve as cross-dataset benchmarks to evaluate out-of-distribution (OOD) generalization capabilities.

**Results.** Table 1 shows the results. Compared to previous MLLM-based methods, our approach achieves superior performance. Meanwhile, compared to score-based methods, our model performs competitively with state-of-the-art (SOTA) methods on KonIQ. Given that Grounding-IQA is not directly trained with the MOS values of KonIQ, this result is impressive. Moreover, in the OOD dataset (KADID and LIVE Challenge), our method achieves the best performance.

In general, the evaluation on score-based IQA tasks further demonstrates the reliability and robustness of our proposed Grounding-IQA in image quality assessment.

## 2 USER STUDY: GIQA-BENCH

We conduct a user study to evaluate more comprehensively. We compare four methods: mPLUG-Owl2, Ferret, Q-Instruct, and our Grounding-IQA.

Table 2: The voting percentages for the three aspects of the user study.

| Method | mPLUG-Owl2-7B (Ye et al., 2024) | Ferret-7B (You et al., 2024a) | Q-Instruct (Wu et al., 2024b) | Grounding-IQA (ours) |
|---|---|---|---|---|
| Response (%) | 6.18 | 6.50 | 36.81 | **50.51** |
| Grounding (%) | N/A | 36.01 | N/A | **63.99** |
| Overall (%) | 1.01 | 12.95 | 6.02 | **80.02** |

Table 3: Performance (PSNR) comparison on image SR ($\times 2$).

| Selection Method | Random | CLIP-IQA (Wang et al., 2023) | Q-Instruct (Wu et al., 2024b) | Grounding-IQA (ours) |
|---|---|---|---|---|
| Urban100 (dB) | 32.75 | 32.82 | 32.84 | **32.89** |
| Manga109 (dB) | 38.92 | 38.96 | 39.97 | **39.00** |

## 2.1 IMPLEMENTATION

For each evaluation, we randomly sample 20 test samples from GIQA-Bench (mixing DES and VQA tasks) and present the results of all four methods in a randomized, anonymized order via a web interface (Fig. 1). Participants evaluate three criteria: response accuracy, grounding accuracy, and overall quality, voting on the best method for each.

## 2.2 RESULTS

We collect responses from 100 users (20 samples per user), and filter out anomalous cases (response times $< 10$s), obtaining 1,877 valid votes. As reported in Tab. 2, our method achieves the best performance across all aspects, with 80.02% ranking highest in overall quality.

## 3 EVALUATION ON THE DOWNSTREAM TASK: IMAGE SR

We validate the effectiveness of grounding-IQA on image super-resolution ($\times 2$).

## 3.1 IMPLEMENTATION

Filter 1,000 images from LSDI (Li et al., 2023) (84,991 images), by four strategies: (1) random sampling, (2) CLIP-IQA (Wang et al., 2023) (score-based), (3) Q-Instruct (Wu et al., 2024b) (MLLM-based), (4) our Grounding-IQA. We train DAT-light (Chen et al., 2023b) (200K iterations) on each subset and evaluate on Urban100 and Manga109. The selection details are as follows:

- **Grounding-IQA**: GIQA-DES to generate descriptions for each image in LSDIR and select the first 1,000 images with "very good" (or "excellent") image quality.

- **Q-Instruct**: uses the same strategy as grounding-IQA.

- **CLIP-IQA**: select the top 1,000 images based on numerical scores.

## 3.2 RESULTS

As shown in Tab. 3, training on subsets selected by different strategies leads to clear differences in downstream performance. Among all baselines, our Grounding-IQA consistently achieves the best results on both Urban100 and Manga109, surpassing score-based (*i.e.*, CLIP-IQA (Wang et al., 2023)) and MLLM-based (*i.e.*, Q-Instruct (Wu et al., 2024b)). These results indicate that incorporating grounding into evaluation can enhance assessment quality.

## 4 PIPELINE DETAILS

We provide more details about the implementation of the annotation pipeline (Sec. 3.2 of the main paper). This contains the question pool of GIQA-EDS, various prompts, and algorithm details.

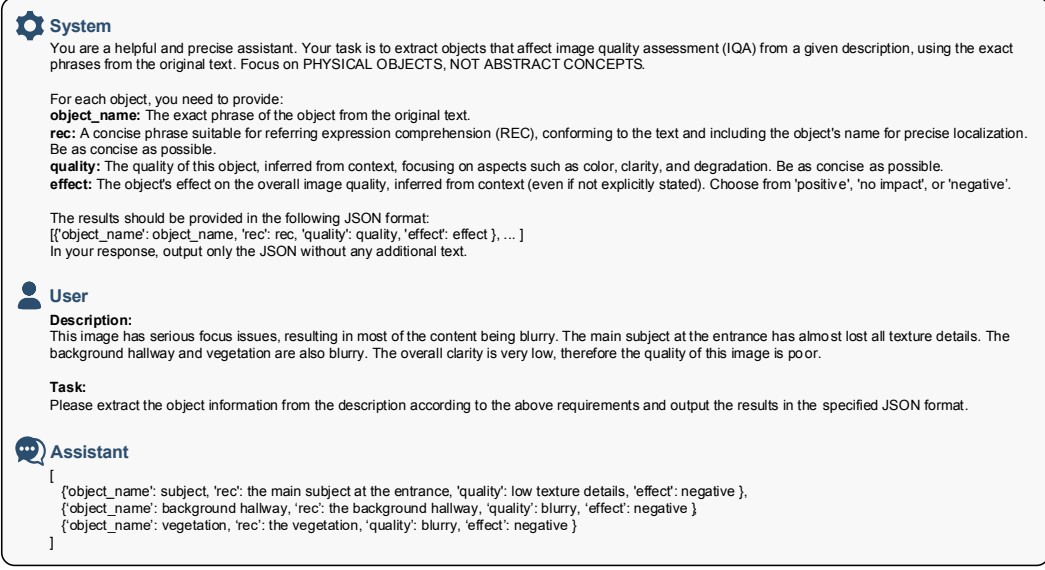

Figure 2: Object Tag Extraction Prompt: Used to extract objects and tags from the description. The instance includes system and user messages and the assistant response. The response is then restructured into the format shown in Fig. 3a (main paper) for subsequent stages.

Table 4: Position Prompt Pool: A set of 15 instructions with varied phrasing, to prompt the model to output precise location information (*i.e.*, bounding box with discrete coordinate).

| No. | Position Prompt | No. | Position Prompt |
|---|---|---|---|
| 1 | Include positional details. | 9 | Include positional information. |
| 2 | Provide locations in the format [label](<x, y>). | 10 | Specify locations in [label](<x, y>) format. |
| 3 | Note positions as [label](<x, y>). | 11 | Reference positions. |
| 4 | Include positional references. | 12 | Cite positions accordingly. |
| 5 | Highlight positions like [label](<x, y>). | 13 | Indicate positions. |
| 6 | Mention positions. | 14 | Include position markers. |
| 7 | Use positional annotations. | 15 | Add positional details. |
| 8 | Note any positions. | | |

## 4.1 GIQA-DES PIPELINE

**Question Pool.** For GIQA-DES, we construct a question pool to generate the $question$ for each data. The GIQA-DES question is divided into two components: task prompt and position prompt. For the **task prompt**, we apply the original question from the source datasets (*i.e.*, Q-Pathway (Wu et al., 2024b) and DQ-495K (You et al., 2024b)), *e.g.*, "Discuss and assess the quality of the picture, and form conclusions based on your evaluation." This helps maintain data consistency. Meanwhile, if no corresponding question exists in source dataset, we use a default question: "Describe and evaluate the quality of the image." For the **position prompt**, we randomly select one from the position prompt pool (Tab. 4). The final GIQA-DES question combines the task prompt and position prompt, *e.g.*, "Describe and evaluate the quality of the image. Note positions as [label](<x, y>)."

**Object Tag Extraction Prompt.** We apply the advanced LLM, *i.e.*, Llama3 (Dubey et al., 2024), to extract key objects and the corresponding tags. The specific prompt is shown in Fig. 2.

**Box Refinement Algorithm Details.** We propose the box refinement algorithm, which consists of two sub-algorithms: IQA-Filter and Box-Merge (Alg. 1, main paper). For clarity, we introduce two functions in more detail: is-touch and coverage-ratio. For **is-touch**, if two patches ($\mathcal{R}[i]$ and $\mathcal{R}[j]$) overlap, is-touch($\mathcal{R}[i], \mathcal{R}[j]$) is is `True`; otherwise, it is `False`. For **coverage-ratio**, the result is:

$$\text{coverage-ratio}(\mathcal{R}[i], \mathcal{R}[j]) = \frac{\text{overlap}(\mathcal{R}[i], \mathcal{R}[j])}{\text{area}(\mathcal{R}[i])}, \tag{2}$$

where overlap($\mathcal{R}[i], \mathcal{R}[j]$) represents the overlap area between two bounding boxes, and area($\mathcal{R}[i]$) denotes the normalized area of the $i$-th bounding box.

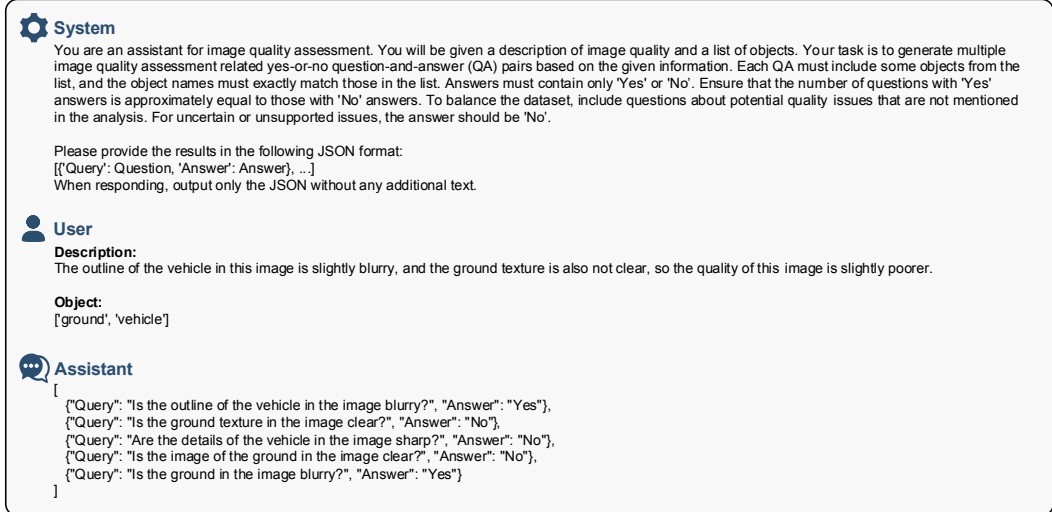

(a) For "Yes/No" questions

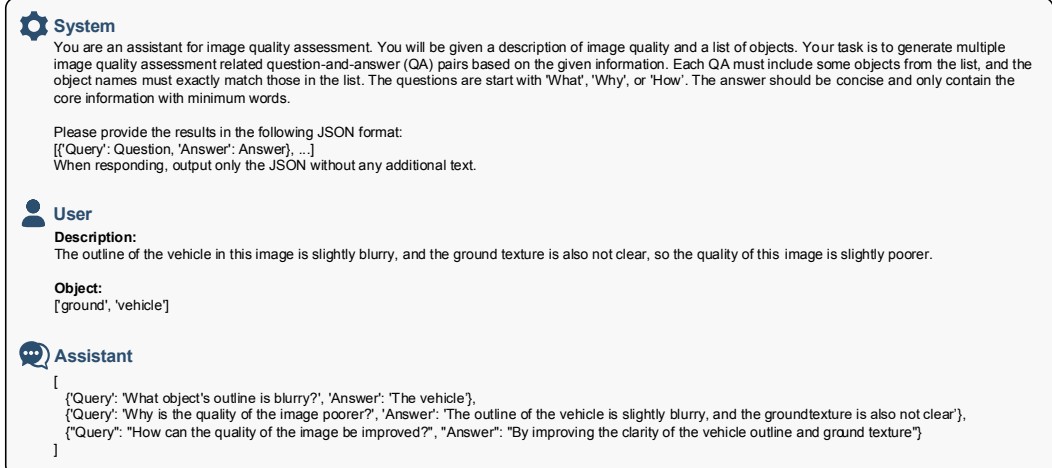

(b) For "What/Why/How" questions

Figure 3: QA Generation Prompt: Applied to generate corresponding QA data from the description and object list. The examples include system and user messages along with the assistant response. For clarity, we omit QA pairs in the response that do not involve the relevant objects. In practical applications, QA pairs that do not contain objects are removed through word matching.

## 4.2 GIQA-VQA PIPELINE

We use the description in GIQA-DES to generate corresponding QA data via an LLM (*i.e.*, Llama3 (Dubey et al., 2024)). We apply two templates to generate two types of questions: "Yes/No" and "What/Why/How". Prompts are shown in Fig. 3.

## 5 METRICS DETAILS

We elaborate on the custom metric defined in Sec. 3.4 of the main paper.

### 5.1 IMPLEMENTATION DETAILS

**LLM-Score.** We introduce the LLM-Score to evaluate description quality across different formats effectively. Given the ground truth and the model-generated description, an LLM (*i.e.*, Llama3 (Dubey et al., 2024)) gives a score ranging from 0 to 4, *i.e.*, 0: "Very Bad", 1: "Bad", 2: "Fair", 3: "Good", 4: "Perfect". The corresponding prompt is illustrated in Fig. 4. The proposed LLM-Score is obtained by scaling the score to 0~100.

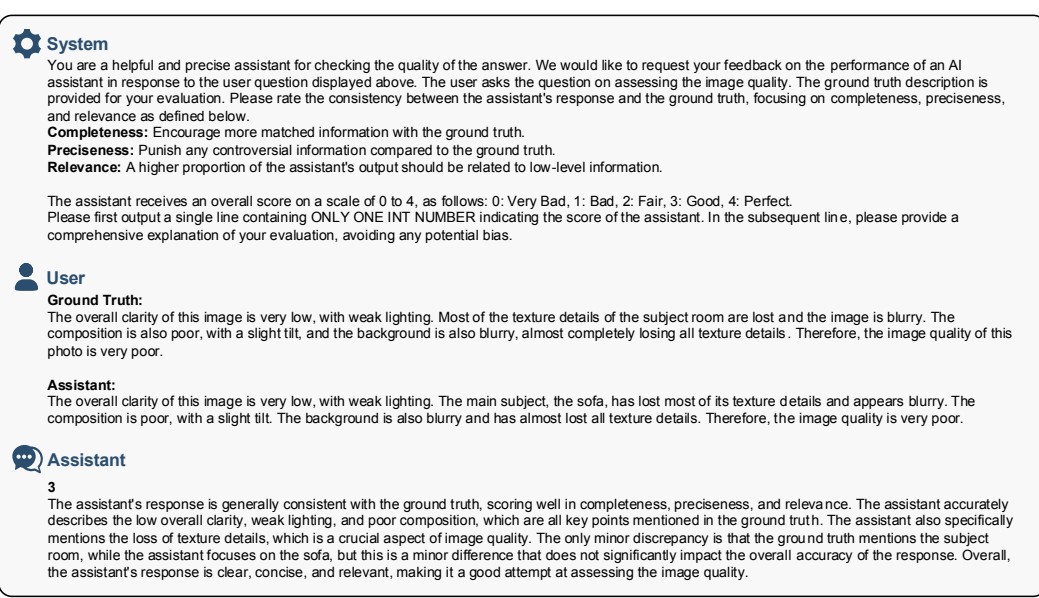

Figure 4: LLM-Score Prompt: Used to assess the quality of the model-generated description in the GIQA-DES task, based on ground truth. The evaluation considers completeness, precision, and relevance, providing an overall score ranging from 0 to 4 (assistant response).

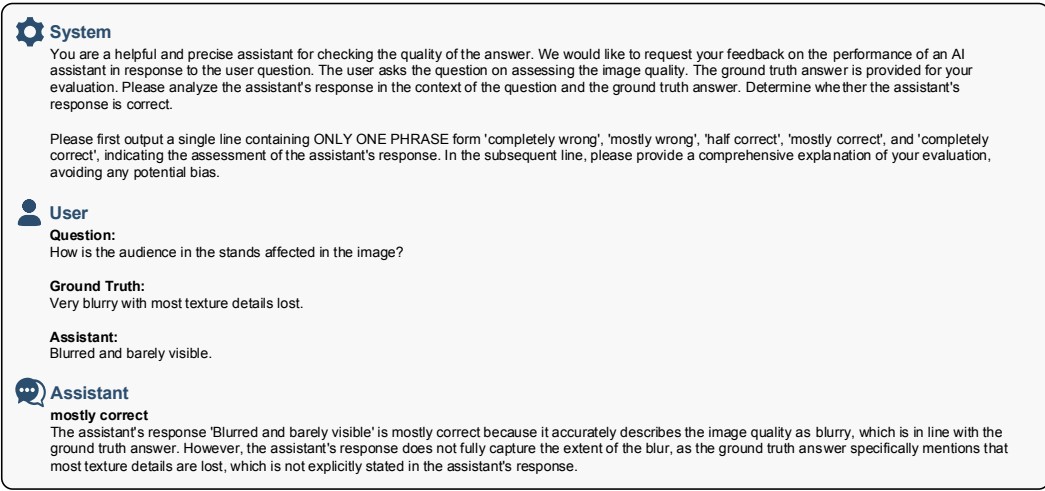

Figure 5: Acc (W) Prompt: Employed to evaluate "What/Why/How" type questions in the GIQA-VQA task. Since direct scoring presents issues, we make the LLM output the corresponding phrase instead. Then, based on the phrase, the score is assigned as: {"completely wrong": 0, "mostly wrong": 1, "half correct": 2, "mostly correct": 3, "completely correct": 4}.

**Acc (W).** For open-ended questions in GIQ-VQA, *i.e.*, "What/Why/How", it is challenging to evaluate correctness by character matching. Thus, we also apply LLM to calculate the accuracy, as shown in Fig. 5. Given the accuracy score of the model response is ranging from 0 to 4, *i.e.*, 0: "completely wrong", 1: "mostly wrong", 2: "half correct", 3: "mostly correct", and 4: "completely correct". The score is then normalized to 0~1 to align with the accuracy scale.

**Tag-Recall.** To evaluate category-specific grounding capabilities, we propose Tag-Recall. First, we calculate the category-agnostic intersection over union (IoU) between each ground truth and generated bounding box. Then, we select pairs with IoU larger than the threshold 0.5, and compute the cosine similarity between the object names associated with the boxes by BERT (Kenton & Toutanova, 2019). If the similarity exceeds 0.5, the pair is considered a true positive (TP). After

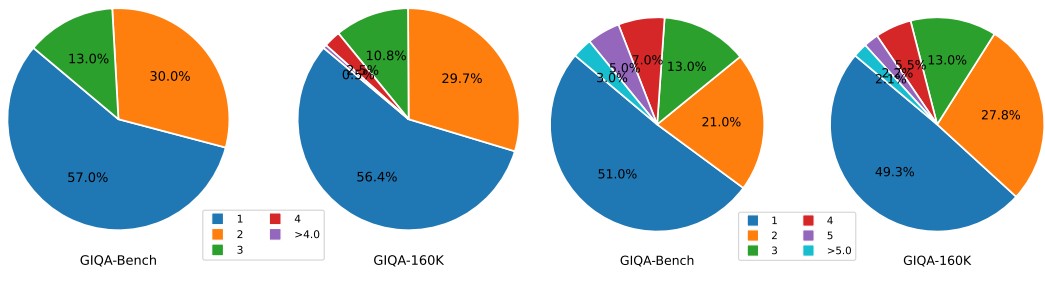

(a) Distribution of object number per data    (b) Distribution of bounding box number per data

Figure 6: More distribution visualization. We visualize the GIQA-DES data distribution in both GIQA-Bench and GIQA-160K, including the number of objects and bounding boxes per data.

marking this pair, we continue detecting TP pairs among the remaining boxes until no further TPs can be identified. The Tag-Recall metric is calculated as:

$$\text{Tag-Recall} = \text{TP}/(\text{TP} + \text{FN}), \tag{3}$$

where FN means the false negative, *i.e.*, the number of missed targets; TP+FN denotes the total number of ground truth boxes. Moreover, for fairness, when calculating Tag-Recall, all bounding boxes are uniformly represented by normalized corner coordinates: $\langle x_1, y_1, x_2, y_2 \rangle$.

## 5.2 METRICS ANALYSES

**Description Quality.** We evaluate the quality of descriptions in GIQA-DES using **BLEU@4** and **LLM-Score**. The BLEU@4 is suitable for cases with similar text paris (*e.g.*, model outputs and ground truth). However, it becomes less accurate when the text is very different. For instance, DepictQA-Wild-7B (You et al., 2024b), despite its good IQA capabilities, achieves a low BLEU@4.

In contrast, LLM-Score can accurately evaluate descriptions with different formats. The DepictQA-Wild-7B achieves much higher LLM-Scores compared to non-IQA methods (*i.e.*, general and ground models). However, LLM-Score may assign relatively high scores to poor descriptions (You et al., 2024b). In such cases, BLEU@4 complements the evaluation.

**VQA Accuracy.** We use accuracy (**Acc**) to evaluate VQA performance. For "Yes/No" questions, word matching provides an accurate assessment. However, for "What/Why/How", while LLM can handle flexible outputs, they lack sufficient distinction in some cases, as analyzed above. This results in relatively small differences in Acc (W) across models. Nevertheless, this metric still reflects the performance, with better models achieving higher Acc (W) scores. In future work, we will explore other evaluation metrics for a more comprehensive analysis.

**Grounding Precision.** We evaluate grounding capability using **IoU** and **Tag-Recall**. The IoU is a category-agnostic metric that measures the quality of generated bounding boxes. This metric evaluates the coverage of the box. However, IoU alone cannot confirm whether the bounding box corresponds to the correct content. Therefore, we apply Tag-Recall to quantify the accuracy of box detection. Combining IoU and Tag-Recall enables a more thorough evaluation.

## 6 MORE DATASET STATISTICS

We provide more statistics for the GIQA-160K and GIQA-Bench datasets. Since GIQA-VQA is derived from GIQA-DES, we focus on GIQA-DES. First, we visualized the distribution of the number of key objects/areas (with coordinates) in each GIQA-DES data within GIQA-Bench and GIQA-160K (Fig. 6a). For GIQA-Bench, which is manually annotated, the single-object description is the most common, accounting for 57.0%. The proportion decreases as the number of objects increases. For GIQA-160K, which is automatically annotated, the distribution is similar to GIQA-Bench, though single-object proportions are slightly higher. This difference may be due to data variations and the limitations of the object detection model. These results also suggest that the number of key objects/areas affecting image quality is generally limited.

Meanwhile, we visualize the bounding boxes number distribution in GIQA-DES (Fig. 6b). Both GIQA-Bench and GIQA-160K show that a single bounding box per description is most common (GIQA-Bench: 51.0%; GIQA-160K: 49.3%). The data proportions also decrease as the number of bounding boxes increases. Moreover, these visualizations indicate that the automatically annotated GIQA-160K exhibits similar distribution as the manually annotated GIQA-Bench.

# 7 GIQA-BENCH COMPARISON DETAILS

We provide more details on comparisons conducted on the GIQA-Bench (Tab. 5 and Fig. 7 in the main paper). It includes fairness analysis, test setting, result analysis, and more qualitative results.

## 7.1 PURPOSE AND FAIRNESS

We explain the purpose of the GIQA-Bench comparisons, along with their fairness considerations.

**Purpose.** The comparisons on the GIQA-Bench aim to demonstrate that existing MLLM methods fine-tuned on our GIQA-160K can achieve superior performance in grounding-IQA tasks. Therefore, we compare three groups of recent methods (*i.e.*, General, Grounding, and IQA). These methods show recent progress on the grounding and IQA tasks. Meanwhile, traditional IQA performance (*e.g.*, score-based IQA) is not the focus of GIQA-Bench. Corresponding results are in Sec. 1.

**Fairness.** Our method introduces a new IQA paradigm, not a novel model architecture. Therefore, fair comparisons should be made under the **same baseline** with **different fine-tuned data**. We provide such fair comparisons on three baselines: LLaVA-v1.5-7B, LLaVA-v1.5-13B, and mPLUG-Owl2-7B. Our Grounding-IQA outperforms the baseline and existing dataset (*e.g.*, Q-Instruct).

Apart from these methods, others (*e.g.*, Ferret (You et al., 2024a) and DepictQA-Wild (You et al., 2024b)) are used to indicate recent progress. Therefore, we directly apply the official pre-trained models without fine-tuning them on the GIQA-160K.

## 7.2 EVALUATION PROMPT DETAILS

We provide detailed information on the comparison methods evaluated on the GIQA-Bench. Overall, for the GIQA-DES task, the query template is: "Describe and evaluate the quality of the image.". For the GIQA-VQA task, the query corresponds to the respective question in the dataset.

**For General.** The general models (General), include LLaVA-v1.5-7B (Liu et al., 2024a), LLaVA-v1.5-13B (Liu et al., 2024a), LLaVA-v1.6-7B (Liu et al., 2024b), and mPLUG-Owl2-7B (Ye et al., 2024). We use the query template defined above for both GIQA-DES and GIQA-VQA. To restrict responses for "Yes/No" questions, we append: "Answer is Yes or No." to the query. Additionally, the image token, *e.g.*, $\langle image \rangle$, is inserted at the query beginning.

**For Ground.** The multimodal referring and grounding models (Ground), contain Shikra-7B (Chen et al., 2023a), Kosmos-2-1.6B (Peng et al., 2024), Ferret-7B (You et al., 2024a), and GroundingGPT-7B (Li et al., 2024). We also apply the query template. And the prompt "Answer is Yes or No." is appended to the end of the corresponding query. Moreover, to implement referring and grounding, we adjust the combination of objects and bounding boxes according to each model definition. Additionally, we make the model explicitly output location coordinates via:

- **Shikra-7B**, we add: "Include the boxes of the items you reference." at the end of the query.
- **Kosmos-2-1.6B**, a special token $\langle grounding \rangle$ is added at the beginning of the query.
- **Ferret-7B**, no additional processing is needed.
- **GroundingGPT-7B**, append "Include object positions in [x0, y0, x1, y1] format".

These settings align with official examples to minimize the impact of prompts on performance.

**For IQA.** The IQA models (IQA), include DepictQA-Wild-7B (You et al., 2024b) and Q-Instruct (Wu et al., 2024b). We use the same query template as above. For DepictQA-Wild, since it lacks specific adjustments for "Yes/No" tasks, we append: "Answer is Yes or No." at the end of the query. For Q-Instruct, we apply the template query without any modifications.

**For Ours.** Our proposed method (*i.e.*, grounding-IQA) directly uses the question corresponding to GIQA-Bench data for both GIQA-DES and GIQA-VQA. All normalized corner coordinates are mapped to discrete coordinates, while other aspects remain unchanged.

### 7.3 MORE ANALYSES ON COMPARISON RESULTS

We provide more analyses of the GIQA-Bench results presented in Sec. 4.3 of the main paper.

**For General.** General models perform poorly on IQA and grounding aspects. Especially, in GIQA-VQA "Yes/No" questions, some models (*e.g.*, LLaVA-v1.5-7B) perform worse than random guessing (*i.e.*, 50%). This may be because the model tends to output "Yes", while "No" answers are more prevalent in GIQA-Bench ("Yes": 35; "No": 55).

**For Ground.** Ground models can realize a certain level of key object grounding. However, their grounding precision (*i.e.*, mIoU and Tag-Recall) is limited due to a lack of low-quality perception. Additionally, while Ferret-7B (You et al., 2024a) performs well on GIQA-DES grounding metrics (*i.e.*, mIoU and Tag-Recall), the visual comparison (Fig. 7 of the main paper) reveals that its results include many objects irrelevant to image quality, which is impractical for real applications.

**For IQA.** IQA models excel in IQA-specific metrics (*i.e.*, BLEU@4 and LLM-Score). Notably, DepictQA-Wild-7B achieves low BLEU@4 scores but higher LLM-Scores than non-IQA methods. This underscores the importance of evaluating IQA performance with multiple metrics to avoid biases. However, existing IQA models lack the ability to realize multimodal referring and grounding.

**For Ours.** General models fine-tuned with our GIQA-160K dataset demonstrate significantly enhanced grounding-IQA capabilities. Compared to models specialized in grounding/referring tasks or IQA tasks, our models achieve better performance in their corresponding applications. It demonstrates the effectiveness of our proposed grounding-IQA.

### 7.4 MORE QUALITATIVE RESULTS

We provide more qualitative results on the GIQA-DES and GIQA-VQA tasks in Figs. 11 and 12. These results supplement the main paper (Fig. 7), further demonstrating the effectiveness of our proposed grounding-IQA. Through multimodal referring and grounding, our grounding-IQA achieves finer-grained assessments, effectively extending the applications of IQA.

## 8 LIMITATIONS AND FUTURE WORK

In this paper, we propose grounding-IQA to achieve a more fine-grained quality assessment. Our current work focuses on no-reference, single-image quality assessment. For reference or multiple image quality assessment, we do not realize. In future work, we plan to extend our method to reference and multiple images to develop more general IQA models. Moreover, developing more efficient and effective annotation pipelines is another direction worth exploring.

## 9 MATERIALS FOR REBUTTAL

In this section, we provide materials relevant to the rebuttal.

### 9.1 VISUALIZATIONS IN MORE SCENARIOS

We further evaluate our method across a broader range of scenarios. The results are shown in Fig. 7. For instance, in the first example, grounding-IQA localizes the text region and provides the

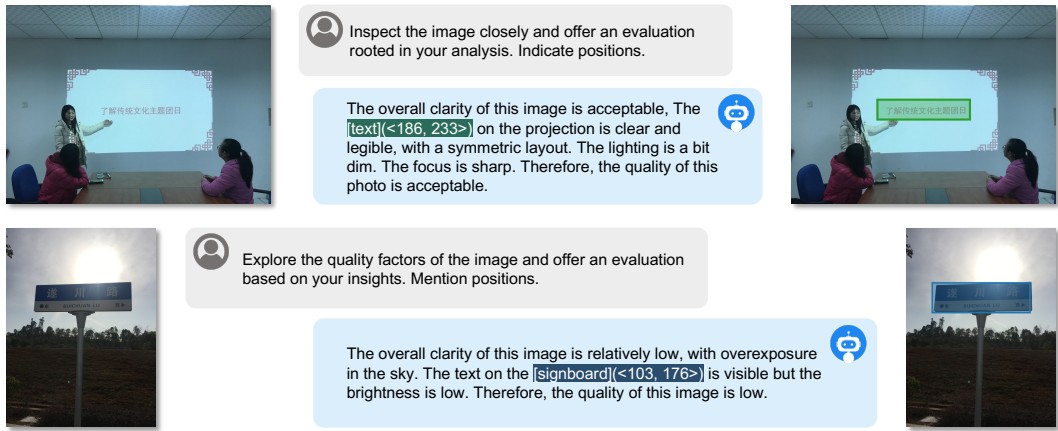

Figure 7: Application in more diverse scenarios (*e.g.*, text). Our Grounding-IQA performs well.

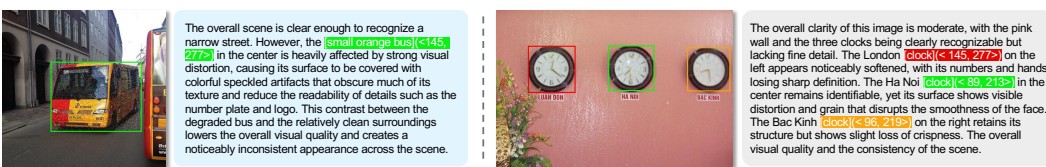

Figure 8: Some instances from the alternative synthetic dataset, Alt-Dataset.

corresponding quality assessment based on it. In the second example, although the text is visible, the lighting is poor, and the sharpness is insufficient. Thus, the overall quality is low. Overall, our method performs well across these diverse cases, demonstrating strong generalization ability.

## 9.2 ALTERNATIVE SYNTHETIC DATASET

**Implementation.** We attempt a synthetic pipeline to construct training data. We build it based on the existing object detection and language understanding dataset RefCOCOg (Yu et al., 2016). RefCOCOg contains 26,711 images and 54,822 segmentation instances (bounding boxes).

Specifically, for each bounding box, we randomly apply one type of degradation: noise, blur, resizing artifacts, JPEG compression, or "unchanged." The degree of degradation is randomly chosen within a specified range. By combining the degraded images, bounding boxes, corresponding Ref-COCOg descriptions, and degradation types&degrees, we apply ChatGPT to generate the final quality descriptions. Given time and resource constraints, we construct **10K** image-text pairs through this process, which we refer to as the **Alt-Dataset**.

**Analysis.** We present some instances in Fig. 8. The synthetic approach provides accurate bounding boxes and controllable degradations. However, synthetic degradations can not reflect the real world well. For example, in the left case, the noise does not match the surrounding environment. Therefore, methods trained on this dataset may perform suboptimally in real-world scenarios.

## 9.3 USER STUDY: IMAGE-DESCRIPTION ALIGNMENT

To evaluate the alignment between the descriptions generated by grounding-IQA and the corresponding images, we conduct a user study.

**Implementation.** For each evaluation, we randomly select 15 samples from GIQA-Bench (GIQA-DES) and present both the images and our descriptions to the users. The users are asked to rate the alignment between the descriptions and the images on a scale from 1 to 5 (higher scores indicate better alignment). The corresponding evaluation page is shown in Fig. 9.

**Results.** We collect feedback from 30 users. The final average score for our method is **4.21**. This indicates that the descriptions generated by our method align well with the images.

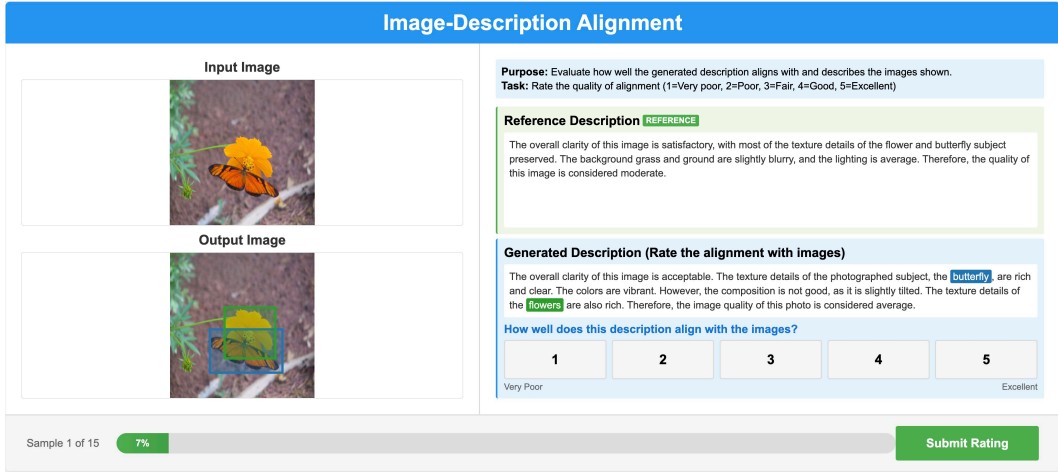

Figure 9: User study page for image-description alignment.

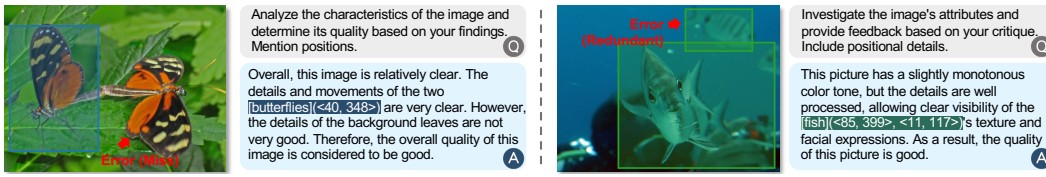

Figure 10: Some failure cases during the annotation process.

### 9.4 FAILURE CASES DURING ANNOTATION

During the data annotation process, some images have low quality or unclear descriptions, and the object detection model used (Grounding DINO (Liu et al., 2024c)) has limitations. Thus, there are some failure cases, as shown in Fig. 10. These failures primarily involve errors in bounding box localization or missing bounding boxes. To address this, we propose the IQA-Filter and Box-Merge algorithms to refine the bounding boxes and eliminate localization errors. Besides, after the automated process, manual verification is performed to remove the remaining incorrect bounding boxes. Finally, from the original **80K** image-text pairs, we construct **66K** grounded pairs (GIQA-DES), removing approximately **14K** low-quality data points.

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

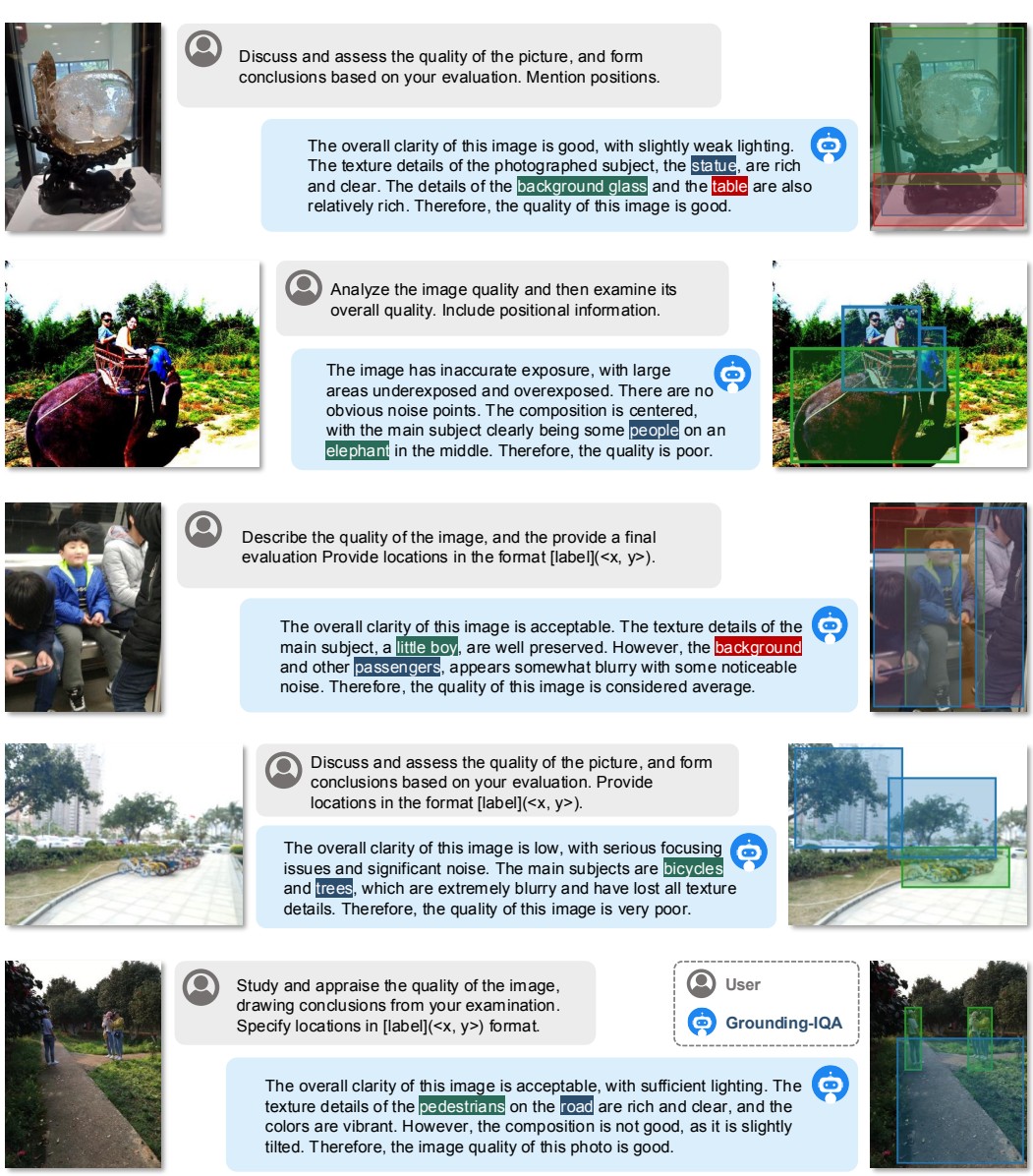

Figure 11: Qualitative results on **GIQA-DES**. For each instance, the left is the input, and the right is the output with bounding boxes. Text highlights correspond with the bounding box colors.

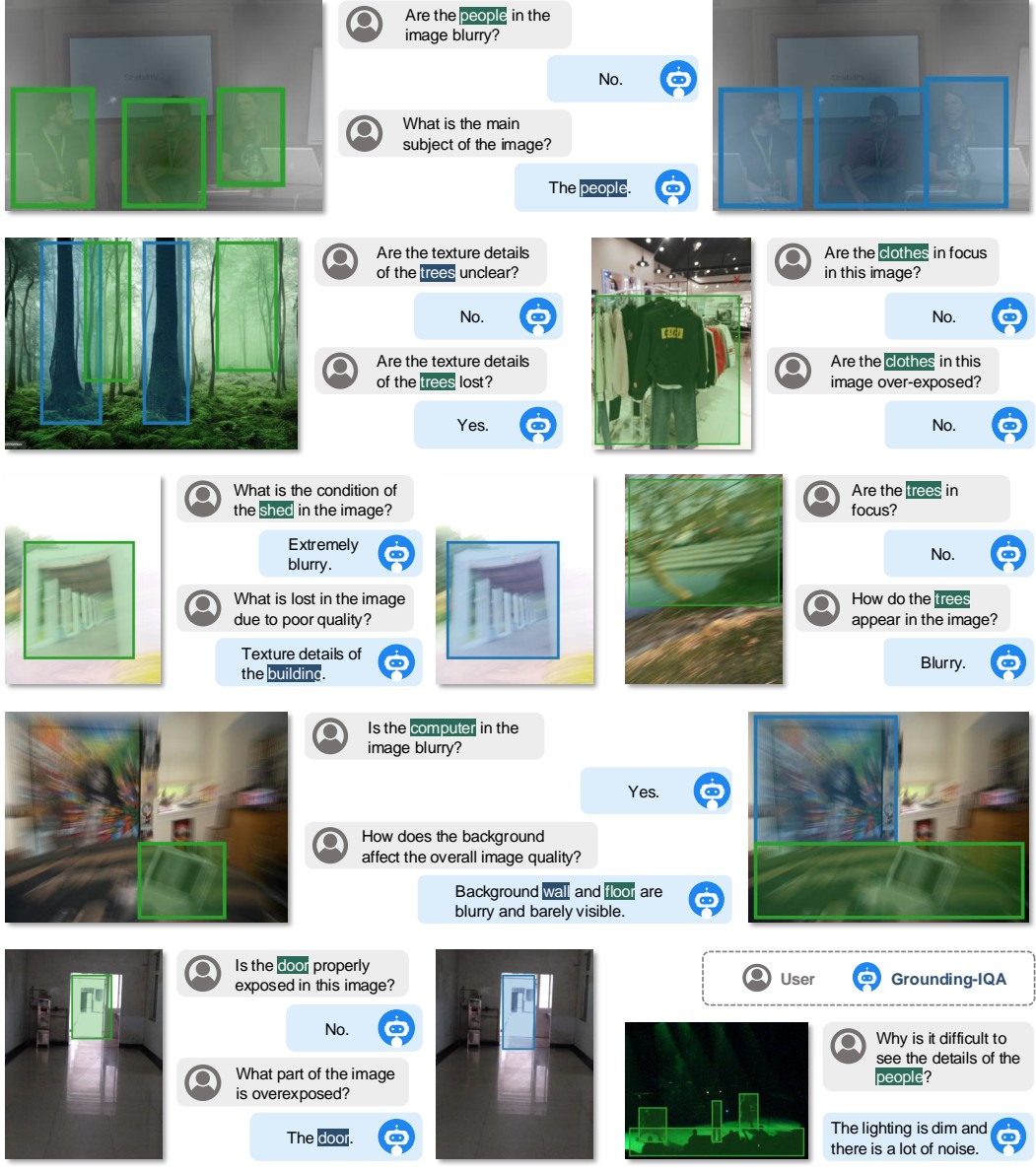

Figure 12: Qualitative results on **GIQA-VQA**. For each instance, the left image is the input with bounding boxes, and the right image (if any) is the visualization of the bounding boxes in the output. Text highlights correspond with the bounding box colors.