# OpenReview forum: "Grounding-IQA: Grounding Multimodal Language Model for Image Quality Assessment"
_ICLR.cc/2026/Conference — ICLR 2026 Poster_

### Official Review · Reviewer_9RGv · 2025-10-24

**Soundness:** 2
**Presentation:** 3
**Contribution:** 2
**Rating:** 2
**Confidence:** 4

**Summary:**

This paper introduces grounding-IQA, a new task paradigm that integrates multimodal referring and grounding capabilities with image quality assessment (IQA) to enable fine-grained quality perception. The authors identify that existing MLLM-based IQA methods primarily rely on general contextual descriptions, limiting their ability to perform detailed quality assessment. To address this, grounding-IQA consists of two subtasks: GIQA-DES (grounding-IQA-description), which provides detailed quality descriptions with precise spatial locations such as bounding boxes, and GIQA-VQA (visual question answering), which focuses on quality-related questions for local image regions. The authors contribute GIQA-160K, a dataset constructed through an automated annotation pipeline, and GIQA-Bench, a comprehensive benchmark that evaluates model performance from three perspectives: description quality, VQA accuracy, and grounding precision. Experimental results demonstrate that the proposed task paradigm, dataset, and benchmark effectively facilitate more fine-grained IQA applications.

**Strengths:**

1.  The paper is well-organized with a clear logical flow, making it easy for readers to understand the motivation, methodology, and contributions. The task paradigm is clearly defined with concrete subtasks (GIQA-DES and GIQA-VQA), facilitating comprehension of the proposed approach.

2.  The proposed automated annotation pipeline for constructing GIQA-160K represents a valuable contribution to the field. This approach addresses the scalability challenge of obtaining fine-grained quality annotations with spatial grounding, potentially benefiting future research in grounded IQA tasks.

3. GIQA-Bench provides a well-designed evaluation framework that assesses model performance from three complementary perspectives: description quality, VQA accuracy, and grounding precision. This multi-faceted evaluation approach enables thorough analysis of grounding-IQA capabilities and sets a solid foundation for future work in this area.

**Weaknesses:**

1. Questionable Task Scope and Definition: The paper claims to address image quality assessment (IQA), which is a broad concept encompassing various quality dimensions such as saturation, color distortion, noise, compression artifacts, etc. However, the proposed method primarily focuses on motion blur of objects, which requires enhanced referring and grounding capabilities. This narrow focus does not adequately represent the full spectrum of IQA. The work would be more convincing if the problem were explicitly defined as "motion blur assessment" or "local distortion grounding" rather than claiming to solve general fine-grained IQA. The current framing creates a mismatch between the stated goal and actual contribution.

2. Limited Generalization Evidence: Even for motion blur assessment, the method's scope appears limited. Most examples showcase salient objects (e.g., people, animals, vehicles), but it remains unclear whether the approach generalizes to other scenarios such as text blur, background element blur, or subtle quality degradations in non-salient regions. The proposed benchmark does not sufficiently demonstrate the model's generalization capability across diverse object categories and blur scenarios, raising concerns about practical applicability.

3. Lack of Evaluation on Public Benchmarks: A critical experimental gap is the absence of evaluation on established public IQA benchmarks. The authors only test their trained model on the self-proposed GIQA-Bench, which limits the ability to assess the method's performance relative to existing approaches and raises questions about potential dataset-specific overfitting. Evaluation on standard IQA datasets would provide stronger evidence of the method's effectiveness and facilitate fair comparison with prior work.

**Questions:**

1. The automated annotation process relies on an existing MLLM (Q-Instruct) to label blur regions, which raises concerns about robustness and error propagation. Can the authors provide evidence demonstrating the reliability of this approach? A potentially more robust alternative would be to synthesize the dataset using pristine images and applying controlled blur degradations at different levels, which would provide natural bounding box annotations and standardized quality labels. How does the current approach compare to such synthetic data generation methods?

2. The paper lacks detailed description of how data contamination between GIQA-160K and GIQA-Bench is prevented. Can the authors explicitly explain the data split strategy and provide evidence (e.g., image similarity checks, source verification) to ensure that the benchmark is completely disjoint from the training data?

3. Given that models like Q-Instruct already possess strong IQA capabilities, wouldn't it be more efficient to enhance their grounding abilities rather than training a new model from scratch? Can the authors justify this design choice and discuss whether a fine-tuning or adapter-based approach on existing IQA-capable MLLMs might achieve comparable or better results with fewer resources?

---

> ### Author Response · Authors · 2025-11-20
> **Response to Reviewer 9RGv (denoted as R4) part 1**
>
> `Q4-1` Questionable Task Scope and Definition: The paper claims to address image quality assessment (IQA), which is a broad concept encompassing various quality dimensions such as saturation, color distortion, noise, compression artifacts, etc. However, the proposed method primarily focuses on motion blur of objects, which requires enhanced referring and grounding capabilities. This narrow focus does not adequately represent the full spectrum of IQA. The work would be more convincing if the problem were explicitly defined as "motion blur assessment" or "local distortion grounding" rather than claiming to solve general fine-grained IQA. The current framing creates a mismatch between the stated goal and actual contribution.
>
> `A4-1` Thank you for the comment.
>
> 1. Our image quality assessment is **not limited to motion blur**. It covers a broad range of quality factors, e.g., sharpness, color, exposure, and noise. We do not specifically constrain our setting to motion blur. In Figs. 2 and 7 of the main paper and Figs. 11 and 12 of the supplementary material, the provided real examples involve diverse quality concepts. For instance, in Fig. 2, the left example shows that Grounding-IQA evaluates object texture, color, and composition.
> 2.  The multimodal grounding is indeed focused on the local region. This design choice aims to enable more fine-grained quality assessment.
> 3. In practice, IQA should consider both **global** and **local** aspects jointly. Many MLLM-based quality description methods focus on global **general** description. Thus, we introduce grounding to strengthen local assessment, thereby achieving more accurate and interpretable fine-grained evaluation.
>
> Overall, we think that describing our task as fine-grained image quality assessment is appropriate and consistent with the actual scope of our method.
>
>
>
> `Q4-2` Limited Generalization Evidence: Even for motion blur assessment, the method's scope appears limited. Most examples showcase salient objects (e.g., people, animals, vehicles), but it remains unclear whether the approach generalizes to other scenarios such as text blur, background element blur, or subtle quality degradations in non-salient regions. The proposed benchmark does not sufficiently demonstrate the model's generalization capability across diverse object categories and blur scenarios, raising concerns about practical applicability.
>
> `A4-2` Thank you for the comment.
>
> We evaluate our method in more scenarios, such as text regions. The corresponding results are provided in **Fig. 7** of the supplementary material. We observe that our approach performs well in these scenarios. It demonstrates the practical applicability of our method.
>
> We will explore applications in a broader range of more diverse scenarios.

---

> ### Author Response · Authors · 2025-11-20
> **Response to Reviewer 9RGv (denoted as R4) part 2**
>
> `Q4-3` Lack of Evaluation on Public Benchmarks: A critical experimental gap is the absence of evaluation on established public IQA benchmarks. The authors only test their trained model on the self-proposed GIQA-Bench, which limits the ability to assess the method's performance relative to existing approaches and raises questions about potential dataset-specific overfitting. Evaluation on standard IQA datasets would provide stronger evidence of the method's effectiveness and facilitate fair comparison with prior work.
>
> `A4-3` Thank you for the comment.
>
> We have evaluated Grounding-IQA on **public, standard score-based IQA datasets** in Sec. 1 (supplementary material), to demonstrate the generalization ability of our method.
>
> **Implementation.**
>
> To apply MLLM-based methods (including Grounding-IQA) to score-based IQA tasks, we adopt the **softmax-based strategy** introduced in Q-Bench. Specifically, we use the following conversational template:
>
> ```shell
> #User: Rate the quality of the image [IMAGE].
> #Assistant: The quality of the image is [SCORE_TOKEN]
> ```
>
> We then compute the IQA score as: $\mathrm{IQA\text{-}Score} = e^{\mathrm{good}} / (e^{\mathrm{good}}+e^{\mathrm{poor}})$,
>
> where $e^{\mathrm{good}}$ and $e^{\mathrm{poor}}$ denote the logits of the terms "good" and "poor" at the position of ([SCORE_TOKEN]).
>
>
>
>
> **Results.**
>
> We conduct experiments on the standard score-based IQA benchmark: **KADID** and **LIVE Challenge**. We compare Grounding-IQA with existing score-based IQA models (HyperIQA, MUSIQ, CLIP-IQA+, Q-Align) as well as MLLM-based approaches (mPLUG-Owl2, DepictQA-Wild, Q-Instruct). The results are shown in the table below.
>
> | Method               |   KADID    |            | LIVE Challenge |            |
> | -------------------- | :--------: | :--------: | :------------: | :--------: |
> |                      |   SRCC ↑   |   PLCC ↑   |     SRCC ↑     |   PLCC ↑   |
> | HyperIQA             |   0.4680   |   0.5060   |     0.7490     |   0.7720   |
> | MUSIQ                |   0.5560   |   0.5750   |     0.8300     |   0.7890   |
> | CLIP-IQA+            |   0.6540   |   0.6530   |     0.8050     |   0.8320   |
> | Q-Align              |   0.6840   |   0.6740   |     0.8600     |   0.8530   |
> | mPLUG-Owl2-7B        |   0.5410   |   0.5460   |     0.4980     |   0.5188   |
> | DepictQA-Wild        |   0.7385   |   0.6816   |     0.5734     |   0.5594   |
> | Q-Instruct           |   0.6980   |   0.6760   |     0.8192     |   0.7981   |
> | Grounding-IQA (ours) | **0.7715** | **0.7648** |   **0.8781**   | **0.8652** |
>
> Grounding-IQA achieves superior performance across these benchmarks. Importantly, Grounding-IQA is not trained directly on score-based IQA tasks, further demonstrating its strong generalization capability. Additional details and results are provided in **Sec. 1 of the supplementary material**.
>
>
>
> Furthermore, we include more experiments to validate the effectiveness of our approach, including the **user study** and **application to the downstream task**. These results further support the generalizability of Grounding-IQA. Details are in the supplementary material (**Sec. 2** and **Sec. 3**).

---

> ### Author Response · Authors · 2025-11-20
> **Response to Reviewer 9RGv (denoted as R4) part 3**
>
> `Q4-4` The automated annotation process relies on an existing MLLM (Q-Instruct) to label blur regions, which raises concerns about robustness and error propagation. Can the authors provide evidence demonstrating the reliability of this approach? A potentially more robust alternative would be to synthesize the dataset using pristine images and applying controlled blur degradations at different levels, which would provide natural bounding box annotations and standardized quality labels. How does the current approach compare to such synthetic data generation methods?
>
> `A4-4` Thank you for the questions.
>
> **For reliable evidence.**
>
> 1. Our method does not rely on the MLLM to annotate blur (or **degradation**) regions. The MLLM is used only for **filtering**. It is easier and more robust than generating annotations from scratch. Specifically, we obtain grounding regions by combining quality-description text, tokenization, and object detection. The MLLM serves as a filter to remove mismatched or low-quality pairs.
> 2. As shown in Tab. 2(a) of the main paper, the MLLM-based refinement step improves the final performance of Grounding-IQA. This proves the reliability of the method.
> 3. Furthermore, we visualize the distribution of our **automatically** generated GIQA-160K dataset compared with the **manually** annotated GIQA-Bench in Fig. 6 in the main paper and Sec. 6 of the supplementary material. Their distributions are similar, indicating that the robustness of the automated pipeline, which performs comparably to human annotation.
>
>
>
> **For the alternative pipeline.**
>
> Following the suggestion of the reviewer, we construct an alternative synthetic dataset as follows:
>
> - We apply RefCOCOg to construct the corresponding dataset. The RefCOCOg contains 26,711 images and 54,822 segmented instances (bounding box).
> - For each bounding box, we apply one of several degradations: noise, blur, resizing artifacts, JPEG compression, or “unchanged.” (**The degradations are not limited to blur to align with our Grounding-IQA task.**)
> - We combine the degraded image, original RefCOCOg description sentence, bounding box, and degradation type, and use ChatGPT to generate the final quality description.
> - Considering resources and time, we construct 10K datasets through this process, which we refer to as the **Alt-Dataset**. Visual examples are shown in **Fig. 8** of the supplementary material.
>
>
>
> We train two versions of the model based on mPLUG-Owl2-7B. One is trained on Alt-Dataset, and one is trained on a 10K subset of GIQA-DES (randomly sampled for fairness). We evaluate both models on the GIQA-Bench DES task, and the results are:
>
> | Method            |   mIoU ↑   | Tag-Recall ↑ | BLEU@4 ↑  | LLM-Score ↑ |
> | ----------------- | :--------: | :----------: | :-------: | :---------: |
> | Alt-Dataset (10K) |   0.4830   |    0.4463    |   18.87   |    55.50    |
> | GIQA-DES (10K)    | **0.5236** |  **0.4968**  | **20.14** |  **61.00**  |

---

> ### Author Response · Authors · 2025-11-20
> **Response to Reviewer 9RGv (denoted as R4) part 4**
>
> `Q4-5` The paper lacks detailed description of how data contamination between GIQA-160K and GIQA-Bench is prevented. Can the authors explicitly explain the data split strategy and provide evidence (e.g., image similarity checks, source verification) to ensure that the benchmark is completely disjoint from the training data?
>
> `A4-5` Thank you for the question. Our GIQA-160K and GIQA-Bench are independent.
>
> 1. The construction process of GIQA-160K and GIQA-Bench inherently ensures that the two datasets are **completely disjoint**:
>
>    - GIQA-160K is generated through our **automated** annotation pipeline.
>    - GIQA-Bench is created by **manual** annotation. We select 100 images of diverse types and quality that do not appear in GIQA-160K. Then we perform multiple rounds of professional manual calibration to obtain the final data.
>    - In other words, GIQA-Bench is not a held-out subset of GIQA-160K. It is independently constructed. The two datasets do not share any information.
>
> 2. To further verify the independence between the two datasets, we perform a similarity analysis.
>
>    Specifically, to simplify computation, we randomly select **1,000 images** from GIQA-160K and compute their similarity to the **100 images** in GIQA-Bench. We extract image features using **ResNet-50** (with normalization) and compute the **pairwise cosine similarity** between all images across the two subsets. We then take the **average** similarity as the dataset-level similarity score.
>
>    The resulting similarity score is **0.1213**, indicating that GIQA-160K and GIQA-Bench are largely dissimilar. This similarity score further confirms that the datasets are well isolated and contain no data contamination.
>
>
>
> `Q4-6` Given that models like Q-Instruct already possess strong IQA capabilities, wouldn't it be more efficient to enhance their grounding abilities rather than training a new model from scratch? Can the authors justify this design choice and discuss whether a fine-tuning or adapter-based approach on existing IQA-capable MLLMs might achieve comparable or better results with fewer resources?
>
> `A4-6` Thank you for the suggestion.
>
> We experiment with training on the Q-Instruct pretrained model (which is built on mPLUG-Owl2-7B). We fine-tune Q-Instruct, and evaluate the resulting models on GIQA-Bench.
>
> | Method                  |   GIQA-DES   |             |   GIQA-VQA   |               |
> | ----------------------- | :----------: | :---------: | :----------: | :-----------: |
> |                         | Tag-Recall ↑ | LLM-Score ↑ | Tag-Recall ↑ | Acc (Total) ↑ |
> | Q-Instruct (fine-tuned) |    0.4870    |    61.00    |    0.4935    |    0.7083     |
> | Grounding-IQA           |  **0.5474**  |  **63.00**  |  **0.7372**  |  **0.7417**   |
>
> We can find that Grounding-IQA trained directly from a general MLLM (mPLUG-Owl2-7B) performs better. This may be because models like Q-Instruct, which are already trained on IQA data, tend to be overfitted to specific IQA patterns, resulting in reduced adaptability and weaker learnability when additional grounding supervision is introduced.
>
> Overall, considering both final performance and trainability, starting from a general-purpose MLLM and training Grounding-IQA directly proves to be a more effective strategy.

---

> ### Author Response · Authors · 2025-11-25
> **Further discussion with Reviewer 9RGv**
>
> Dear Reviewer 9RGv,
>
> We sincerely appreciate the time you dedicate to reviewing our paper and providing your valuable feedback. We have carefully considered your comments and provided detailed responses:
>
> 1. We clarify the **task scope** and **problem definition**.
> 2. We validate our method in broader application scenarios, such as text-related cases.
> 3. We report evaluation results on **public score-based IQA** benchmarks.
> 4. We analyze the **robustness** of the automated annotation pipeline and experiment with the **alternative** construction strategy you suggested.
> 5. We explain the **separation** between GIQA-160K and GIQA-Bench and provide supporting evidence.
> 6. We conduct experiments using IQA-pretrained MLLMs (e.g., **Q-Instruct**).
>
> As the author–reviewer discussion phase is nearing its end, we would like to confirm whether our responses address your concerns. If any questions or suggestions remain, please feel free to let us know. We would be grateful to hear them.
>
>
>
> Thank you again for your time and valuable feedback.
>
> Best regards,
>
> Authors

---

> ### Author Response · Authors · 2025-11-27
> **Further discussion with Reviewer 9RGv (second follow-up)**
>
> Dear Reviewer 9RGv,
>
> We sincerely appreciate your valuable time and thoughtful feedback on our submission. As the discussion period is nearing its end, we would like to kindly follow up to check whether our previous responses have addressed your concerns.
>
> If there are any remaining issues or points that would benefit from further clarification, please let us know. We would be glad to provide additional information.
>
> Thank you once again for your time and consideration.
>
> Best regards,
>
> Authors

---

> > ### Comment · Reviewer_9RGv · 2025-11-27
> > **feedback from reviewer**
> >
> > Thanks for the rebuttals of the authors, which have mostly addressed my concerns. I d like to raise my scores to 6.

---

> > > ### Author Response · Authors · 2025-11-27
> > > **Thanks Reviewer 9RGv for approving our work**
> > >
> > > Dear Reviewer 9RGv,
> > >
> > > We are glad to receive your response and are pleased that our replies address your concerns. We will adopt your suggestions in the revised version.
> > >
> > > Best regards,
> > >
> > > Authors

---

### Official Review · Reviewer_hAUf · 2025-10-26

**Soundness:** 4
**Presentation:** 3
**Contribution:** 3
**Rating:** 8
**Confidence:** 5

**Summary:**

This paper proposes a new IQA paradigm named Grounding-IQA. By combining multimodal referring/grounding with IQA, the framework achieves more fine-grained image quality evaluation. The authors design two tasks, GIQA-DES and GIQA-VQA, and construct an automated annotation pipeline to build a dataset named GIQA-160K. The authors also propose GIQA-Bench for performance evaluation. Both quantitative and qualitative results demonstrate the effectiveness of the proposed Grounding-IQA.

**Strengths:**

1. Introducing multimodal grounding into IQA is a reasonable idea that aligns with human assessment logic. It also improves interpretability.
2. The dataset GIQA-160K and the benchmark GIQA-Bench are well-designed contributions to the community. The automated pipeline is well-structured, which supports further research.
3. Both quantitative and qualitative results validate the effectiveness of the proposed Grounding-IQA. Comparisons with various MLLMs (general, grounding, IQA) further confirm its advantages.
4. Additional experiments in the supplementary material, including traditional score-based IQA tasks, user studies, and downstream tasks, provide further evidence of effectiveness.
5. Ablation studies and visualizations support the validity of the designed pipeline and dataset.
6. The paper is well written, with extensive figures, tables, and pseudocode that facilitate understanding.
7. The supplementary material includes detailed implementation information, visualizations, and analytical results, making the content comprehensive.

**Weaknesses:**

1. Although implementation details of the pipeline are provided, some settings lack explanation or analysis. For example, why the discrete coordinates are set to 20×20, and why the number of tasks differs in GIQA-Bench?
2. The work applies Llama-3 for evaluation. Considering the existence of more advanced models such as Llama-4, using them could provide more accurate evaluation results.
3. The impact of dataset scale on performance is not investigated. It is not certain whether 160K data is necessary.

**Questions:**

1. Explain the considerations of specific parameter settings in the pipeline.
2. Evaluate LLM-Score and Acc (W) using Llama-4 and analyze the results.
3. Conduct ablation studies on the dataset (GIQA-160K) size.

---

> ### Author Response · Authors · 2025-11-20
> **Response to Reviewer hAUf (denoted as R3)**
>
> `Q3-1` Although implementation details of the pipeline are provided, some settings lack explanation or analysis. For example, why the discrete coordinates are set to 20×20, and why the number of tasks differs in GIQA-Bench?
>
> `A3-1` Thank you for the suggestion.
>
> **For the discrete coordinates.** We conduct an ablation study on the coordinate resolution $(m, n)$, where $m=n=10, 20, 40$. The results are evaluated on GIQA-DES, shown in the table below.
>
> | $(m, n)$ |   mIoU ↑   | Tag-Recall ↑ | BLEU@4 ↑  | LLM-Score ↑ |
> | -------- | :--------: | :----------: | :-------: | :---------: |
> | (10, 10) |   0.5369   |    0.5264    |   23.08   |    61.75    |
> | (20, 20) | **0.5851** |  **0.5497**  | **23.67** |  **61.75**  |
> | (40, 40) |   0.5447   |    0.4672    |   23.06   |    61.50    |
>
> When the number of discrete bins is small (e.g., 10), the predicted bounding boxes become coarse, which limits the grounding capability. In contrast, using a larger number of bins (e.g., 40) increases the difficulty of representation learning and degrades performance. Therefore, we choose $m = n = 20$ to balance localization precision and learning difficulty.
>
>
>
> **For the number of tasks in GIQA-Bench.**
>
> GIQA-VQA contains two variants (VQA-Y and VQA-W). Thus, we increase the total number of VQA tasks slightly to ensure a more comprehensive evaluation.
>
>
>
> `Q3-2` The work applies Llama-3 for evaluation. Considering the existence of more advanced models such as Llama-4, using them could provide more accurate evaluation results.
>
> `A3-2` Thank you for the comment.
>
> We attempt to evaluate our method using a more advanced LLM for calculating the LLM-Score and Acc (W). Since we do not have access to Llama-4 due to licensing restrictions, we instead adopt another publicly available model, Qwen3. The results are shown below:
>
> | Method              | Q-Instruct | Grounding-IQA |     △      |
> | ------------------- | :--------: | :-----------: | :--------: |
> | LLM-Score (Llama-3) |   62.00    |     63.00     |    1.00    |
> | Acc (W) (Llama-3)   |   0.5375   |    0.5875     |   0.0500   |
> | LLM-Score (Qwen3)   |   61.50    |     63.25     |  **1.75**  |
> | Acc (W) (Qwen3)     |   0.5208   |    0.5833     | **0.0625** |
>
> We observe that under the better LLM, our Grounding-IQA achieves larger improvements the over existing approach (Q-Instruct), which aligns well with the visual results. Besides, the relative ordering of the scores across different LLMs remains consistent, demonstrating the robustness of the evaluation method.
>
>
>
> `Q3-3` The impact of dataset scale on performance is not investigated. It is not certain whether 160K data is necessary.
>
> `A3-3`
>
> Thank you for the suggestion. We conduct experiments using datasets of different scales, and the results are shown below:
>
> | Number |   mIoU ↑   | Tag-Recall ↑ | BLEU@4 ↑  | LLM-Score ↑ |
> | ------ | :--------: | :----------: | :-------: | :---------: |
> | 40K    |   0.5366   |    0.5093    |   20.92   |    62.50    |
> | 80K    |   0.5771   |    0.5203    |   22.22   |    62.75    |
> | 160K   | **0.5955** |  **0.5474**  | **22.87** |  **63.00**  |
>
> We observe that larger training sets lead to better performance. This demonstrates the importance of using the full 160K samples.
>
>
>
> `Q3-4` Explain the considerations of specific parameter settings in the pipeline.
>
> `A3-3` Thank you for the comment. We set the discrete coordinate resolution $(m, n)$ to 20 to balance localization precision and learning difficulty.
>
> Besides, we slightly increase the number of VQA tasks in GIQA-Bench to enable more comprehensive evaluation.
>
> The detailed explanations are provided in `A3-1`.
>
>
>
> `Q3-5` Evaluate LLM-Score and Acc (W) using Llama-4 and analyze the results.
>
> `A3-5` Thank you for the suggestion. Since Llama-4 is not accessible to us, we use Qwen3 as an alternative.
>
> The results show that the Qwen3 has a more reasonable score. On the other hand, the LLM-based scores are robust across different models. See details in `A3-2`.
>
>
>
> `Q3-6` Conduct ablation studies on the dataset (GIQA-160K) size.
>
> `A3-6` Thank you for the advice.
>
> We conduct ablation studies on the size of GIQA-160K. The results show that larger datasets lead to better performance, indicating that using the total 160K samples is necessary. The detailed results are provided in `A3-3`.

---

### Official Review · Reviewer_WkTJ · 2025-10-27

**Soundness:** 4
**Presentation:** 3
**Contribution:** 4
**Rating:** 8
**Confidence:** 5

**Summary:**

The paper proposes Grounding-IQA, a new paradigm for Image Quality Assessment (IQA) that couples spatial grounding with quality reasoning so models can locate and describe the local regions responsible for perceived degradations. It introduces two tasks: GIQA-DES (quality description with precise locations) and GIQA-VQA (region-aware quality QA where questions/answers include coordinates). To train and evaluate, the authors build GIQA-160K via an automatic pipeline and a 100-image evaluation set GIQA-Bench that scores description quality, VQA accuracy, and localization. Fine-tuning several mainstream MLLMs on GIQA-160K yields consistent gains on GIQA-Bench over generic MLLMs, pure grounding models, and pure IQA models, indicating better fine-grained perceptual quality understanding.

**Strengths:**

1. The paper integrates grounding with IQA to localize the regions causing quality defects—making it highly actionable for editing and restoration.
2. Its end-to-end labeling/training pipeline—detection, IQA filtering, box merging, and coordinate–text fusion—is scalable and reproducible.
3. A large training corpus and a targeted benchmark with multi-faceted metrics (generation, QA, localization) enable comprehensive evaluation.
4. Consistent gains across multiple MLLM backbones, with ablations indicating the effectiveness of multi-task training and box-handling strategies.

**Weaknesses:**

1. Fix typographical errors in the references.
2. Some novel IQA methods like (VisualQuality-R1) should be included.
3. Conduct user studies to evaluate how well the output descriptions align with the corresponding images.

**Questions:**

See weakness.

---

> ### Author Response · Authors · 2025-11-20
> **Response to Reviewer WkTJ (denoted as R2)**
>
> `Q2-1` Fix typographical errors in the references.
>
> `A2-1` Thank you for your comment. We have corrected the typographical errors.
>
>
>
> `Q2-2` Some novel IQA methods like (VisualQuality-R1) should be included.
>
> `A2-2` Thank you for the suggestion. We include comparisons with the new IQA method, VisualQuality-R1.
>
> VisualQuality-R1 is a score-based NR-IQA model. Therefore, we compare it with our method on score-based IQA benchmarks (KonIQ, KADID, and LIVE Challenge). The results are shown below:
>
> | Method              | LIVE Challenge |            |   KonIQ    |            |
> | ------------------- | :------------: | :--------: | :--------: | :--------: |
> |                     |     SRCC ↑     |   PLCC ↑   |   SRCC ↑   |   PLCC ↑   |
> | VisualQuality-R1    |     0.8762     | **0.8940** |   0.8946   |   0.9060   |
> | Grouding-IQA (ours) |   **0.8781**   |   0.8652   | **0.9342** | **0.9282** |
>
> Our method achieves comparable performance to VisualQuality-R1. Notably, our model is not trained specifically for score-based IQA. It demonstrates the strong generalization ability of our method in quality assessment.
>
>
>
> `Q2-3` Conduct user studies to evaluate how well the output descriptions align with the corresponding images.
>
> `A2-3` Thank you for the advice. We conduct a user study to evaluate the alignment between the output descriptions and the corresponding images.
>
> **Setup:** For each evaluation round, we randomly sample 15 groups from GIQA-Bench (GIQA-DES) and present the images along with our Grouding-IQA generated descriptions to the users. Users are asked to rate the consistency between the output description and the image on a 1~5 scale (higher is better). The evaluation web is provided in the supplementary material (**Fig. 9**).
>
> **Results:** We collect feedback from 20 users, and the final average score of our method is **4.21**.
>
> This indicates that the descriptions generated by our method align well with the images, demonstrating the effectiveness of our approach.

---

> > ### Comment · Reviewer_WkTJ · 2025-11-25
> >
> > Thanks for the rebuttal. I think this work presents an elegant application of IQA. Its fine‑grained distortion detection could be very useful for other tasks—like visual generation—as a specific reward model. So I’ll keep my rating and recommend the paper for acceptance.

---

> > > ### Author Response · Authors · 2025-11-25
> > > **Thanks Reviewer WkTJ for approving our work**
> > >
> > > Dear Reviewer WkTJ,
> > >
> > > We are glad to receive your response. We sincerely appreciate your positive acknowledgement of our response and our work. We will continue to further explore the applications of Grounding-IQA in future research.
> > >
> > > Best regards,
> > >
> > > Authors

---

### Official Review · Reviewer_U2TN · 2025-11-01

**Soundness:** 3
**Presentation:** 2
**Contribution:** 2
**Rating:** 4
**Confidence:** 3

**Summary:**

The paper introduces Grounding-IQA, a novel Image Quality Assessment (IQA) approach tailored for multimodal large model learning (MLLM). The framework decomposes IQA into two subtasks: (1) GIQA-DES, which performs fine-grained, subject-level quality evaluation via grounding-based descriptions, and (2) GIQA-VQA, which assesses low-level attributes at localized regions using a visual question answering paradigm. The authors further present GIQA-160K, a large-scale dataset constructed with an automated annotation pipeline, and GIQA-Bench, an evaluation benchmark that jointly measures description quality, VQA accuracy, and grounding precision.

**Strengths:**

The manuscript is clearly organized, with a strong logical flow and well-designed figures and tables that effectively illustrate the framework and experimental setup. The writing quality is generally high, facilitating easy comprehension of complex ideas. Furthermore, the benchmark construction is comprehensive and well-motivated — evaluating models from multiple dimensions (descriptive, interrogative, and spatial grounding) offers a holistic view of multimodal IQA performance.

**Weaknesses:**

While the contribution of a large-scale dataset and benchmark is substantial, the main novelty resides primarily in dataset creation and annotation procedures rather than in methodological innovation. The proposed framework mainly adapts existing MLLM capabilities and fine-tuning strategies to the IQA domain. As such, the work may align more closely with a dataset or benchmark track rather than the ICLR main track, where new learning methodologies or theoretical insights are typically emphasized.
Additionally, there are a few minor grammatical issues throughout the text—for example, “provide” should be “provides” (line 53), and “new coordinates is” should be “new coordinates are” (line 268).

**Questions:**

1.Dataset Construction Details：The paper mentions that GIQA-160K was built through an “automated annotation pipeline.” Could the authors elaborate on how annotation quality and consistency were ensured? Was there any human verification or filtering process?How are noisy or ambiguous quality descriptions handled? Are there examples of failure cases in the annotation process?
2.Whether human evaluation was included to validate these metrics.Without this information, it’s difficult to judge how faithfully GIQA-Bench reflects perceptual IQA quality.
The framework decomposes into GIQA-DES and GIQA-VQA. How much does each component individually contribute to overall IQA performance? An ablation study showing the relative importance of each subtask (and their potential synergy) would clarify whether the decomposition is truly beneficial or merely conceptual.
4.It would be constructive if the authors could discuss potential limitations — for instance, the reliance on large-scale multimodal models that are computationally expensive to train and deploy, or the dataset’s possible bias toward certain types of degradation. A discussion on how future work might reduce these dependencies or enhance scalability would improve the paper’s outlook.

---

> ### Author Response · Authors · 2025-11-20
> **Response to Reviewer U2TN (denoted as R1) part 2**
>
> `Q1-4` Whether human evaluation was included to validate these metrics.Without this information, it’s difficult to judge how faithfully GIQA-Bench reflects perceptual IQA quality.
>
> `A1-4` Thank you for the questions. We conduct the corresponding manual assessment. Specifically:
>
> 1. Our GIQA-Bench evaluates performance from three perspectives: description quality, VQA accuracy, and grounding precision.
> 2. Metrics such as mIoU, Tag-Recall, BLEU@4, and Acc (Y) are widely used and well-established evaluation measures. We adopt them to assess different aspects of IQA performance.
> 3. In addition, we introduce **LLM-Score**, which uses LLM to generate **description** quality scores. To ensure interpretability, we carefully design the prompts to enable the LLM to provide both a score and an explanation. We **manually** evaluate and iteratively **optimize** the prompts by assessing 50 question-answer pairs to ensure consistency and reasonableness between the generated scores and explanations.
> 4. Furthermore, we conduct a **user study** to evaluate the description quality and grounding ability of different methods in Sec. 2 (supplementary material). The user study shows that Grounding-IQA > Q-Instruct >> Ferret & mPLUG-Owl2 in terms of description quality. This **relative ranking** is consistent with the LLM-Score results, demonstrating the effectiveness and validity of our evaluation metric (LLM-Score).
>
>
>
> `Q1-5` The framework decomposes into GIQA-DES and GIQA-VQA. How much does each component individually contribute to overall IQA performance? An ablation study showing the relative importance of each subtask (and their potential synergy) would clarify whether the decomposition is truly beneficial or merely conceptual.
>
> `A1-5` Thank you for the comment. We conduct an ablation study on each subtask in Tab.3 of the main paper. We apply mPLUG-Owl2-7B as the baseline. The results are provided below:
>
> | Method   |   GIQA-DES   |             |   GIQA-VQA   |               |
> | -------- | :----------: | :---------: | :----------: | :-----------: |
> |          | Tag-Recall ↑ | LLM-Score ↑ | Tag-Recall ↑ | Acc (Total) ↑ |
> | Baseline |     N/A      |    48.25    |     N/A      |    0.5633     |
> | Only-DES |  **0.5497**  |    61.75    |    0.5577    |    0.5900     |
> | Only-VQA |    0.3283    |    38.50    |    0.4872    |    0.7217     |
> | DES+VQA  |    0.5474    |  **63.00**  |  **0.7372**  |  **0.7417**   |
>
> From the results, we observe:
>
> - **GIQA-DES alone** improves the baseline in terms of description quality and grounding precision, but its QA ability remains limited.
> - **GIQA-VQA alone** yields marginal gains in grounding performance, likely due to the lack of contextual information.
> - In contrast, **jointly training on both GIQA-DES and GIQA-VQA** leads to substantial improvements across all dimensions.
>
> These findings demonstrate that the task decomposition and the multi-task training strategy are effective, providing complementary benefits rather than only serving as a conceptual design.
>
>
>
> `Q1-6` It would be constructive if the authors could discuss potential limitations — for instance, the reliance on large-scale multimodal models that are computationally expensive to train and deploy, or the dataset’s possible bias toward certain types of degradation. A discussion on how future work might reduce these dependencies or enhance scalability would improve the paper’s outlook.
>
> `A1-6` Thank you for the suggestion. In Sec. 8 (supplementary material), we already discuss limitations and feature works related to settings (NR and FR) and the efficiency of the annotation pipeline. Below, we further elaborate on additional potential limitations and future directions.
>
> 1. Since MLLMs rely on large-scale data, our grounding-based IQA framework applies extensive fine-tuning. However, this large-scale training introduces a large overhead. A promising direction for future work is to explore more efficient fine-tuning strategies, such as incorporating explicit grounding supervision or improving model architectures to enhance grounding accuracy with fewer resources.
> 2. Our dataset currently relies on degradations present in existing datasets, which limits control over the range and diversity of degradation types. A future direction is to introduce synthetic degradations to construct more diverse and controllable training data, thereby improving performance across degradation scenarios.

---

> ### Author Response · Authors · 2025-11-20
> **Response to Reviewer U2TN (denoted as R1) part 1**
>
> `Q1-1` While the contribution of a large-scale dataset and benchmark is substantial, the main novelty resides primarily in dataset creation and annotation procedures rather than in methodological innovation. The proposed framework mainly adapts existing MLLM capabilities and fine-tuning strategies to the IQA domain. As such, the work may align more closely with a dataset or benchmark track rather than the ICLR main track, where new learning methodologies or theoretical insights are typically emphasized.
>
> `A1-1` Thank you for your comment.
>
> 1. Our core contribution is **framework (mechanism) innovation**. We introduce multimodal grounding into the IQA task, establishing a new **paradigm**, Grouding-IQA. The novel paradigm realizes fine-grained quality assessment.
> 2. To realize the paradigm, we design two grounding-based **sub-tasks,** GIQA-DES and GIQA-VQA. Furthermore, we develop the corresponding dataset, pipeline, and benchmark. The dataset enables fine-grained, grounding-aware IQA. The dataset and pipeline are part of the overall innovation of our framework, designed to implement our new paradigm, Grounding-IQA.
> 3. Introducing a new paradigm, rather than model architecture design, is a common practice in the IQA community. Previous works, e.g., AesExpert (ACMMM), Q-Bench (ICLR), and Q-Instruct (CVPR), are of this type.
>
>
>
> `Q1-2` Additionally, there are a few minor grammatical issues throughout the text—for example, “provide” should be “provides” (line 53), and “new coordinates is” should be “new coordinates are” (line 268).
>
> `A1-2` Thank you for the suggestion.
>
> We review the manuscript and correct the grammatical issues.
>
>
>
> `Q1-3` Dataset Construction Details: The paper mentions that GIQA-160K was built through an “automated annotation pipeline.” Could the authors elaborate on how annotation quality and consistency were ensured? Was there any human verification or filtering process? How are noisy or ambiguous quality descriptions handled? Are there examples of failure cases in the annotation process?
>
> `A1-3` Thank you for the questions.
>
> 1. In our pipeline, we ensure annotation quality through the Stage-3 (box refinement) process, where the **IQA-Filter** and **Box-Merge** algorithms remove erroneous annotations and refine bounding boxes.
> 2. After completing the automated pipeline, we perform manual verification to remove image-text pairs with incorrect bounding boxes.
> 3. As a result, from the raw 80K image-text pairs, we obtain 66K grounded pairs (GIQA-DES). Approximately 14K of low-quality data are removed.
> 4. These procedures effectively ensure data quality. In Fig. 6 (main paper) and Sec. 6 (supplementary material), we visualize the distribution of the **automatically** generated training data (GIQA-VQA160K) and the **manually** annotated test data (GIQA-Bench). The two distributions are highly similar, demonstrating the effectiveness of our automated construction process.
> 5. **Failure cases**: Some failure cases occur during annotation, mainly incorrect bounding boxes. We provide examples in **Fig. 10** (supplementary material). But through Stage-3 refinement and manual filtering, we successfully remove these incorrect samples.

---

> ### Comment · Area_Chair_B3jb · 2025-11-21
> **Reviews are generated by LLM**
>
> Dear reviewers and authors,
>
> I have received a reminder indicating that this review report may have been generated by an LLM. After examining the review, I indeed found that the reviewer did not approach this manuscript with a responsible attitude (as is evident from the formatting). I must say that this is one of the sloppiest review reports I have ever seen—the reviewer did not even bother to include basic line breaks. I have taken note of this issue and will address it appropriately when making the final decision. This reviewer will also be marked as a very low-quality reviewer.

---

> ### Author Response · Authors · 2025-11-25
> **Further discussion with Reviewer U2TN**
>
> Dear Review U2TN,
>
>
>
> We sincerely thank you for the time and thoughtful consideration you devote to reviewing our paper. We have carefully considered your comments and provided detailed responses to your questions:
>
> 1. We clarify our core contribution: **framework (mechanism) innovation**.
> 2. We explain and demonstrate the quality and consistency of the **dataset annotation**.
> 3. We analyze the effectiveness of the **evaluation metrics**.
> 4. We provide evidence for the **roles and contributions** of the DES and VQA subtasks.
> 5. We review the manuscript and discuss additional **limitations and future work**.
>
> As the author-reviewer discussion phase is drawing to a close, we would like to confirm whether our responses adequately address your concerns. If you have any further questions, please feel free to let us know. We would be grateful to hear them.
>
> Thank you again for your time and thoughtful feedback.
>
> Best regards,
>
> Authors

---

> ### Author Response · Authors · 2025-11-27
> **Further discussion with Reviewer U2TN (second follow-up)**
>
> Dear Reviewer U2TN,
>
> We sincerely appreciate the time and thought you have already devoted to reviewing our paper. As the author–reviewer discussion period is approaching its conclusion, we would like to kindly follow up again regarding your feedback.
>
> We hope to discuss further with you whether or not your concerns have been addressed. If you have any additional questions or if further clarification would be helpful, please let us know.
>
> Thank you again for your time and consideration.
>
> Best regards,
> Authors

---

### Author Response · Authors · 2025-11-25
**Response to all reviewers and area chairs for a brief summary**

Dear reviewers and area chairs,



We sincerely thank all reviewers (**R1-U2TN**, **R2-WkTJ**, **R3-hAUf**, **R4-9RGv**) and the area chairs for your time and valuable comments.

We are pleased to note that:

1. **R2, R3, and R4** recognize our proposed paradigm (grounding-IQA), the definition of the associated subtasks, and **R2 and R3** acknowledge the practical value of our method.
2. **R2 and R4** acknowledge the importance and scalability of our data annotation pipeline.
3. **All reviewers** agree that our benchmark provides a comprehensive evaluation of grounding-IQA.

We have responded to each reviewer individually, and we summarize the key points below:

1. We **clarify the framework innovation** of our method.
2. We provide analyses demonstrating the **consistency of the dataset** and the **validity of our evaluation metrics**.
3. We perform experiments on the **decomposition of the subtasks**.
4. We include comparisons to the novel IQA method, **VisualQuality-R1**.
5. We conduct a **user study** to evaluate image-description alignment.
6. We add more **ablation** experiments, including pipeline parameters, evaluation with a more advanced LLM, and dataset-scale studies.
7. We clarify the **task scope and definition**.
8. We evaluate our method in **broader scenarios**, such as text-related cases.
9. We provide evaluation results on public **score-based IQA** benchmarks.
10. We analyze the **robustness** of the annotation pipeline and attempt an **alternative** construction strategy.
11. We present evidence confirming the **complete separation** of GIQA-160K and GIQA-Bench.
12. We evaluate the impact of fine-tuning **IQA-pretrained** MLLMs for our task.



We again thank all reviewers and the area chairs.

Best regards,

Authors

---

### Author Response · Authors · 2025-11-29
**Rebuttal-Discussion Summary and Reviewer Feedback Status**

Dear Program Chairs, Senior Area Chairs, Area Chairs, and Reviewers:

We sincerely thank all program chairs, senior area chairs, area chairs, and reviewers (**R1-U2TN**, **R2-WkTJ**, **R3-hAUf**, **R4-9RGv**) for the time and expertise dedicated to evaluating our work. To facilitate the final decision, we provide a concise summary of the rebuttal discussion and the current status of reviewer feedback.

### Summary of Rebuttal-Discussion

Overall, the majority of reviewers recognize the core contributions of this paper, including:

1. The proposed **Grounding-IQA paradigm** and its clear task formulation.
2. The scalable and important data annotation **pipeline**.
3. The comprehensive **benchmarking** provides practical value for IQA research.

During the discussion phase, we addressed all raised concerns, including:

1. **Clarifying** our framework innovation, dataset consistency, metric reliability, subtask effectiveness, task scope, annotation robustness, and train–test separation.
2. Adding more **experiments**: comparisons to a new IQA method, conducting further user studies, performing more ablation experiments, evaluating broader scenarios, exploring alternative annotation strategies, and testing models pretrained for IQA.
3. We **re-provide** the results on public score-based IQA benchmarks, the user study on grounding-IQA, and subtask ablation to address the reviewers' concerns better.

### Current Status of Reviewer Feedback

1. **R2-WkTJ** and **R3-hAUf** consistently provide positive assessments, acknowledging the clarity and practical impact of our contributions, and further confirm this after discussion.
2. **R4-9RGv** explicitly states that their concerns are resolved and indicates a **positive revision to a score of 6**.
3. **R1-U2TN:** We have provided clear explanations and experiments for all concerns raised by this reviewer. This should resolve their issues. Since the reviewer did not engage in the discussion phase, we think the initial score may not reflect the actual situation and should be reconsidered accordingly.

### Overall

Our paper makes novel contributions in task paradigm, data pipeline, and evaluation design, and we validate the effectiveness and scalability of our approach through extensive experiments. Except for the reviewer (U2TN), who did not participate in the discussion, the opinions **of the reviewers overall** converge toward a **positive consensus**.

We sincerely hope that the final decision may take these observations into careful consideration.

Thank you again for your time, constructive feedback, and support.

Best regards,

Authors

---

### Meta-Review · Area_Chair_1Mns · 2025-12-27

**Summary:**

The paper introduces a novel IQA paradigm, its core novelty lies in integrating multimodal referring/grounding with image quality assessment, consisting of two subtasks (GIQA-DES and GIQA-VQA). It constructs the GIQA-160K dataset via an automated annotation pipeline and the GIQA-Bench benchmark, enabling more fine-grained image quality evaluation.

Key concerns—such as comparison with more advanced models, demonstration of the IQA method’s generalization, and introduction of a user study—were adequately addressed during the rebuttal. Overall, the work is easy to follow and the experiments are comprehensive. I recommend accepting this paper and think this work is a strong fit for the ***"Datasets and Benchmarks"*** track.
Thank you for flagging U2TN. This feedback has been taken into account in the decision.

**Reviewer Concerns:**

All concerns were adequately addressed.

**Reviewer Scores:**

Reviewer 9RGv may raise the score.

---

### Decision · Program_Chairs · 2026-01-26

Accept (Poster)